# The human gut and groundwater harbor non-photosynthetic bacteria belonging to a new candidate phylum sibling to Cyanobacteria

Sara C Di Rienzi[1†], Itai Sharon[2†], Kelly C Wrighton[2], Omry Koren[1], Laura A Hug[2], Brian C Thomas[2], Julia K Goodrich[1], Jordana T Bell[3], Timothy D Spector[3], Jillian F Banfield[2,4]*, Ruth E Ley[1]*

[1]Department of Microbiology, Cornell University, Ithaca, United States; [2]Department of Earth and Planetary Science, University of California, Berkeley, Berkeley, United States; [3]Department of Twin Research and Genetic Epidemiology, King's College London, London, United Kingdom; [4]Department of Environmental Science, Policy, and Management, University of California, Berkeley, Berkeley, United States

**Abstract** Cyanobacteria were responsible for the oxygenation of the ancient atmosphere; however, the evolution of this phylum is enigmatic, as relatives have not been characterized. Here we use whole genome reconstruction of human fecal and subsurface aquifer metagenomic samples to obtain complete genomes for members of a new candidate phylum sibling to Cyanobacteria, for which we propose the designation 'Melainabacteria'. Metabolic analysis suggests that the ancestors to both lineages were non-photosynthetic, anaerobic, motile, and obligately fermentative. Cyanobacterial light sensing may have been facilitated by regulators present in the ancestor of these lineages. The subsurface organism has the capacity for nitrogen fixation using a nitrogenase distinct from that in Cyanobacteria, suggesting nitrogen fixation evolved separately in the two lineages. We hypothesize that Cyanobacteria split from Melainabacteria prior or due to the acquisition of oxygenic photosynthesis. Melainabacteria remained in anoxic zones and differentiated by niche adaptation, including for symbiosis in the mammalian gut.

*For correspondence:
jbanfield@berkeley.edu (JFB);
rel222@cornell.edu (REL)

†These authors contributed equally to this work

Competing interests: The authors declare that no competing interests exist.

## Introduction

Among the geochemical changes that have occurred over the past few billion years, perhaps the most dramatic was the transformation of the Earth's atmosphere and upper oceans into oxygen-rich environments (*Bekker et al., 2004*). Cyanobacteria are presumed responsible for this geochemical revolution, as they comprise the sole lineage known to have innovated the production of oxygen as a byproduct of photosynthesis (*Mulkidjanian et al., 2006*). Further, nitrogen fixation by Cyanobacteria is central to the Earth's nitrogen cycle (*Vitousek et al., 2002*). Cyanobacteria are inferred to be one of the earliest branching bacterial lineages (*Altermann and Kazmierczak, 2003*; *Bekker et al., 2004*) and have diversified across environments—land, fresh, and salt water, and all levels of the photic zone (*Dworkin, 2006*). Via endosymbiosis, Cyanobacteria became the chloroplasts of plants (*Sagan, 1967*), a role that underlines their broad evolutionary importance.

The evolutionary history of the Cyanobacteria phylum is only partially resolved: no related taxa, from which a common ancestor could be inferred, have been described. However, recent culture-independent 16S rRNA gene surveys of microbial communities have revealed a novel clade sibling and basal to Cyanobacteria (*Ley et al., 2005*). The presence of organisms related to Cyanobacteria in the

**eLife digest** Microbes are ubiquitous in the world and exist in complex communities called microbiomes that have colonized many environments, including the human gut. Until modern techniques for sequencing nucleic acids became available, many of the organisms found in these microbiomes could not be studied because they could not be cultured in the laboratory. However, advances in sequencing technology have made it possible to study the evolution and properties of these microbes, including their impact on human health.

Bacteria belonging to the phylum Cyanobacteria had a significant effect on the prehistoric Earth because they were the first organisms to produce gaseous oxygen as a byproduct of photosynthesis, and thus shaped the Earth's oxygen-rich atmosphere. Early plants took up these bacteria in a symbiotic relationship, and plastids—the organelles in plant cells that perform photosynthesis and produce oxygen–are the descendants of Cyanobacteria.

Organisms evolutionarily related to Cyanobacteria have been found in the human gut and in various aquatic sources, but these bacteria have not been studied because it has not been possible to isolate or culture them. Now, Di Rienzi, Sharon et al. have used modern sequencing techniques to obtain complete genomes for some of these bacteria, which they assign to a new phylum called Melainabacteria.

By analyzing these genomes, Di Rienzi, Sharon et al. were able to make predictions about the cell structure and metabolic abilities of Melainabacteria. Like Cyanobacteria, they have two membranes surrounding the bacterial cell; unlike Cyanobacteria, however, they have flagella that propel them through liquid or across surfaces. Most interestingly, Melainabacteria are not able to perform photosynthesis, but instead produce energy through fermentation and release hydrogen gas that can be consumed by other microorganisms.

The genome of the bacteria isolated from water reveals that it has the capacity to fix nitrogen. Cyanobacteria can also fix atmospheric nitrogen, but the protein complexes used by the two phyla are not related, which suggests that nitrogen fixation evolved after the evolutionary divergence of Cyanobacteria and Melainabacteria.

By exploring previously published datasets of bacterial communities, Di Rienzi, Sharon et al. found that Melainabacteria are common in aquatic habitats. They are also prevalent in the guts of herbivorous mammals and humans with a predominantly vegetarian diet. Melainabacteria from the human gut also synthesize several B and K vitamins, which suggests that these bacteria are beneficial to their host because in addition to aiding with the digestion of plant fibers, they are also a source of vitamins.

gut is notable because they are widely shared across individuals (*Consortium HMP, 2012*), where members of this group can comprise up to 20% of the total sequences recovered from stool (*Dethlefsen and Relman, 2011*), as well as shared across various mammalian species (*Ley et al., 2008*). As Cyanobacteria are photosynthetic organisms, it has been assumed that these sequences represent genomic material derived from ingestion of chloroplasts or Cyanobacterial cells (*Turnbaugh et al., 2009*; *Koenig et al., 2011*; *Consortium HMP, 2012*). However, given their large evolutionary separation from Cyanobacteria and the lack of cultured representatives, no conclusion as to the roles of these predominant organisms of the human gut has been possible.

Related bacteria deep-branching from the Cyanobacteria have also been detected in water and other anoxic environments including sediments (*Ley et al., 2005*). An earlier phylogenetic reconstruction based on full-length 16S rRNA gene sequences indicated that this water-soil-sediment-derived clade is distinct from that made up entirely of gut-derived sequences (*Ley et al., 2005*). This mapping of habitats onto the two clades suggested that niche adaptation had shaped the groups' evolution and hence their phylogeny, but beyond this observation the lineage remained enigmatic.

New sequencing methods and bioinformatics advances provide a route for genomic analysis of uncultured organisms from complex microbial communities (*Dick et al., 2009*; *Iverson et al., 2012*; *Wrighton et al., 2012*). We show that these methods can yield complete and near-complete genomes from relatively low abundance organisms, without the need for single cell genomic approaches. Here, we analyze eight curated genomes from bacteria from intestinal samples and an aquifer sediment to

evaluate their metabolisms and roles in their respective habitats. The analysis provides clues as to the ancestral state of the lineage that gave rise to these organisms and to the Cyanobacteria.

## Results

### Reconstruction of genomes from metagenomic samples

We generated metagenomes from three fecal samples obtained from three healthy adult humans (termed A, B, and C; *Table 1*). In addition, we identified genome fragments that derived from an organism deeply branching with respect to Cyanobacteria in a microbial community metagenomic dataset from a subsurface aquifer (*Wrighton et al., 2012*). The microbial communities in the human fecal samples and the subsurface differ substantially. The bacterial communities of human fecal samples A, B, and C are typical of human fecal microbiota, as they are predominantly composed of members of the Firmicutes and Bacteroidetes, with a few other phyla represented (*Figure 1A–C*) (*Consortium HMP, 2012*). The subsurface sample, on the other hand, has a greater phylum-level phylogenetic diversity with the most abundant members belonging to Proteobacteria and the candidate phyla OD1 and OP11 (*Figure 1D*). For both sample types, the abundances of genomes from the Cyanobacteria sibling clade were less than 5% of the total community.

Despite the relatively low abundance of these genomes in the samples (*Table 1*), recently developed algorithms that improve the assembly and manual curation of metagenomic data (*Sharon et al., 2013*) allowed us to recover two genomes from sample A (MEL.A1, MEL.A2), two from sample B (MEL.B1, MEL.B2), and three genomes from sample C (MEL.C1, MEL.C2, MEL.C3) for a total of seven distinct genomes reconstructed from human fecal samples (*Tables 1 and 2*).

Through genome curation, we were able to establish linkage among all scaffolds for four of these genomes (complete genomes; *Table 2*). Completeness was confirmed by validating assembly graph connectivity, and also by considering expected genome features such as single copy genes. Correctness was confirmed by re-assembly of potentially mis-assembled regions such as scaffold ends, and by considering the 'phylogenetic profile' of genes in each scaffold. Our curation method verified unique paired read placement throughout the reconstructed genomes, a requirement consistent with standard methods of isolate genomics. All scaffolds identified as deriving from an organism with some similarity to Cyanobacteria, based on the phylogenetic profile of the encoded genes, were incorporated into the closed, complete genomes. Additional small scaffolds were identified and incorporated using paired read placement. The phylogenetic signal for novelty was robust, because essentially all other genomic fragments (excluding phage and plasmids) shared high similarity with genomes of previously sequenced organisms.

The assembled genomes range from 1.9 to 2.3 Mbp and encode 1,800 to 2,230 genes. Additionally, we analyzed the binned genome, hereafter, ACD20, (*Tables 1 and 2*) from the aquifer dataset (*Wrighton et al., 2012*). The ACD20 genome is larger than the genomes recovered from fecal samples—3.0 Mbp encoding 2,819 genes. Additional genome details are provided in *Tables 1 and 2*. We used all eight genomes for phylogenetic analyses and four representative genomes (three from the gut plus the sediment genome) for the metabolic analyses that follow.

### A new candidate phylum sibling to Cyanobacteria

Corroborating earlier findings (*Ley et al., 2005*), a 16S rRNA gene sequence-based phylogeny built with publically available sequences places the unknown lineages, represented in part by the gut and

**Table 1.** Samples from which Melainabacteria genomes were recovered

| Sample | Environment | # of reads | Abundance% | # genomes recovered |
|--------|-------------|------------|------------|---------------------|
| A | Gut | 109,557,616 | 4 | 1 Complete, 1 Partial |
| B | Gut | 124,163,248 | 3 | 2 Complete |
| C | Gut | 112,578,264 | 2 | 1 Complete, 1 Near Complete, 1 Partial |
| ACD | Aquifer | 232,878,979 | 0.7 | 1 Near Complete |

Sample, number of reads sequenced, and estimates of the abundance of Melainabacteria in the communities based on 16S rRNA gene survey and coverage information. ACD20 was assembled from three samples (see *Wrighton et al., 2012*).

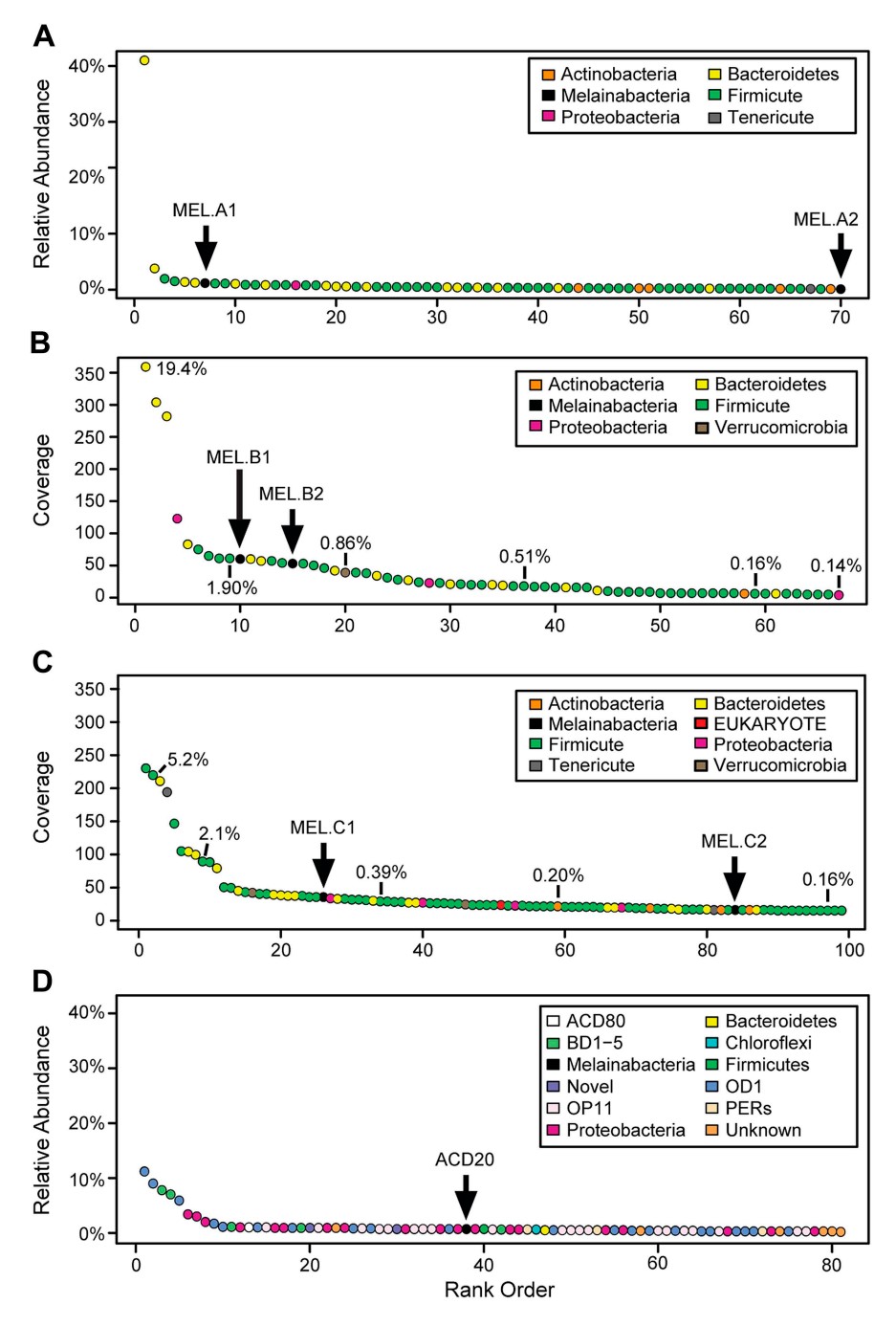

**Figure 1**. Community composition of samples containing Melainabacteria. (**A–C**) The relative composition of the human fecal samples A, B, and C, and (**D**) the aquifer community members. In **A** and **D** estimated percent relative abundance of the community is plotted, and in **B** and **C**, coverage is plotted, but estimated percent relative abundance is noted on the figure for select members. Organisms are classified at the phylum level. The human fecal sample A community is dominated by *Prevotella copri* DSM 18205, which accounts for more than 40% of the sequencing reads and is represented by several strains. Sequencing depth was not sufficient for human fecal sample C to accurately estimate roughly 25% of the community abundance, which includes MEL.C3. Aspects of the community composition of the aquifer sample are discussed in *Wrighton et al. (2012)*.

**Table 2.** Melainabacteria genomes recovered in this study

| Sample ID | Coverage | Genome status | Size (bp) | %GC | Scaffolds | N50 | Coding features | 16S rRNA genes |
|---|---|---|---|---|---|---|---|---|
| ACD20 | 30x | Near Complete | 2,979,548 | 33.5 | 191 | 33,361 | 2,819 | ND |
| MEL.A1 | 73x | Complete | 1,867,336 | 32.9 | 1 | 1,867,336 | 1,832 | 2 |
| MEL.A2 | 5.5x | Partial | 1,192,455 | 30.6 | 88 | 16,613 | 1,386 | ND |
| MEL.B1 | 62x | Complete | 2,302,307 | 35.3 | 21 | 542,117 | 2,219 | 2 |
| MEL.B2 | 44x | Complete | 2,308,205 | 36.3 | 26 | 375,376 | 2,222 | 2 |
| MEL.C1 | 26.5x | Complete | 2,053,642 | 34.1 | 4 | 1,742,055 | 2,120 | 2 |
| MEL.C2 | 27.5x | Near Complete | 2,159,327 | 35.3 | 34 | 146,232 | 2,104 | 2 |
| MEL.C3 | 6x | Partial | 1,323,478 | 29.9 | 93 | 15,878 | 1,472 | ND |

ND = not determined. See the section *Genome assembly* in 'Materials and methods' for an explanation of Genome Status.

aquifer bacteria, basal to photosynthetic Cyanobacteria (*Figure 2A*). Three subgroups are revealed, one of which comprises sequences obtained from animal guts (*Figure 2B*). The 16S rRNA gene sequences of the gut and aquifer bacteria share no more than 84% identity to Cyanobacterial sequences, consistent with placement of these organisms in a new candidate phylum (<85% identity, as suggested previously [*Hugenholtz et al., 1998*]). The bacterial tree has been described as a polytomy due to the inability of 16S rRNA gene phylogenies to capture any specific branching order for the phyla (*Pace, 1997*), so this phylum is unusual in its robustly supported relationship to Cyanobacteria. To further substantiate this evolutionary relationship, we constructed a phylogeny of concatenated ribosomal protein sequences (*Figure 3A,B*). The result shows that the eight new genomes form a monophyletic lineage that branches deeply from the Cyanobacterial lineage, with ACD20 basal to the group (*Figure 3B*). Importantly, a common ancestor for these organisms and photosynthetic Cyanobacteria is well supported (100/100 bootstrap bipartitions) in both trees. With the sum of this evidence, we designate these bacteria as the new candidate phylum Melainabacteria, where 'melaina' refers to the Greek nymph of dark waters.

The newly described melainabacterial genomes contain other genes that reinforce an ancestry shared with Cyanobacteria. For instance, the genomes encode the 30S ribosomal protein S1 *rpsA* gene, rather than the homolog *ypfD*, which is exclusive to the Firmicutes (*Danchin, 2009*). Moreover, two of the gut genomes (MEL.B1 and MEL.B2) encode the A type of RnpB (E-values 5.676e-59, 2.235e-60 respectively), which is found in all Bacteria except the Firmicutes and Tenericutes (*Haas et al., 1996*; *Zwieb et al., 2011*). Three of the four complete genomes also have homologs to the S-layer like COG, CyOG00138 (e.g., *Anabaena variabilis*: Q3MBT3), a protein found only in Cyanobacteria (*Mulkidjanian et al., 2006*). Although these genes are not exclusive to Cyanobacteria, they are important phylogenetic markers, support a shared ancestry with Cyanobacteria, and refute an ancestry with the Firmicutes, a possibility that arises when considering the metabolic genes of the Melainabacteria genomes (see below).

## Melainabacteria are non-photosynthetic and non-respiratory but contain homologs to genes encoding light response regulators in photosynthetic Cyanobacteria

Oxygenic photosynthesis is perhaps the most exceptional characteristic of Cyanobacteria, as all other bacterial photosyntheses are anoxygenic. The Melainabacteria genomes appear to entirely lack genes for photosynthesis (photosystem I and photosystem II, thylakoid membranes, succinate dehydrogenase, and the cytochrome $b_6f$ complex), indicating that none have the capacity for oxygenic or anoxygenic photosynthesis. Also absent are genes for soluble (e.g., plastocyanin or photosynthetic ferredoxin) and membrane-affiliated electron carrier proteins (e.g., cytochromes, quinones, Fe-S, or flavin). Additionally, we found no genes for aerobic respiration including terminal quinol and cytochrome *c* oxidases, terminal reductases involved in anaerobic respiration (e.g., fumarate, nitrate), or carbon fixation pathways. Together these findings suggest that these members of the Melainabacteria are not capable of phototrophy or respiratory metabolism.

Despite being non-photosynthetic, the melainabacterial genomes encode homologs of the circadian rhythm regulators RpaA and RpaB and the high intensity light sensor NblS. The histidine kinase NblS

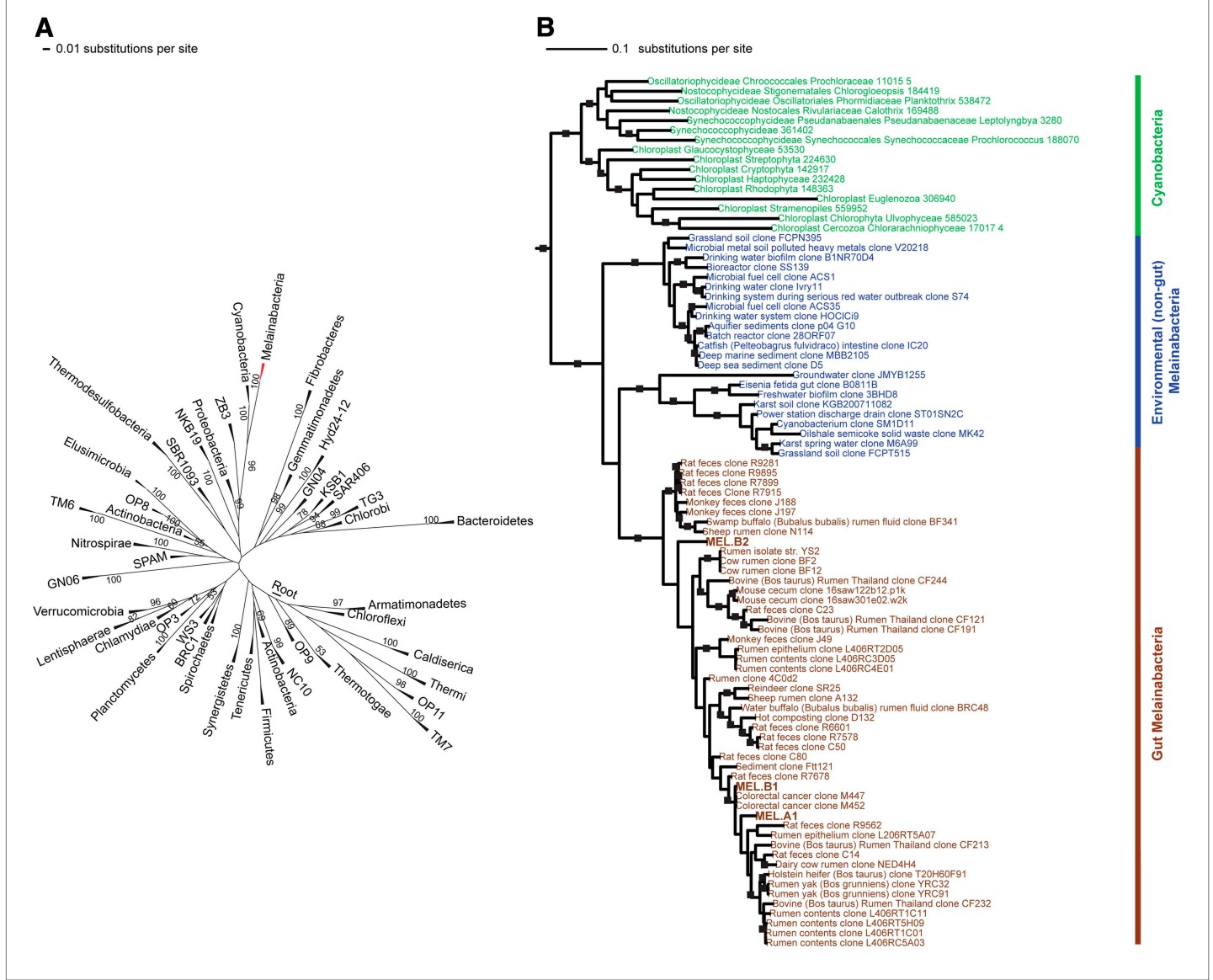

**Figure 2**. 16S rRNA gene phylogeny of Melainabacteria and Cyanobacteria. Trees were built using 16S rRNA gene sequences from MEL.A1, MEL.B1, and MEL.B2 (the 16S rRNA sequence of ACD20 was not recovered). (**A**) 16S rRNA gene phylogeny tree with five representative sequences from each phylum obtained from the Greengenes May 2011 database (**DeSantis et al., 2006**). Bootstrap values greater than 50% are indicated. (**B**) 16S rRNA gene phylogeny built using one representative sequence from each order within Cyanobacteria from the Greengenes database (May 2011) (**DeSantis et al., 2006**) besides orders YS2, SM1D11, and mle1-12 from which all sequences were used. For Melainabacteria, the habitats from which the sequences were predominantly derived are indicated and colored according to isolation source (blue = environmental (non-gut); brown = gut). Cyanobacteria are displayed in green. Bootstrap values greater than 70% are indicated by a black square.

in Cyanobacteria preserves the photosynthetic machinery by regulating its expression and degradation under high and blue/UV-A conditions (**van Waasbergen et al., 2002**). RpaA and RpaB are found in all Cyanobacteria (**Mulkidjanian et al., 2006**), where they regulate the circadian clock KaiABC (**Hanaoka et al., 2012**) and link energy transfer between the antennae and the photosystem (**Ashby and Mullineaux, 1999**). The four complete genomes of Melainabacteria lack SasA, which typically functions as the sensor to the response regulators RpaA and RpaB (**Hanaoka et al., 2012**), as well as any photosynthetic machinery, which suggests that these proteins may have another function. Similarly, the NblS homolog in the gut and aquifer Melainabacteria most likely has a different function. The current KEGG database (**Kanehisa et al., 2012**) indicates that RpaA and NblS are exclusive to Cyanobacteria.

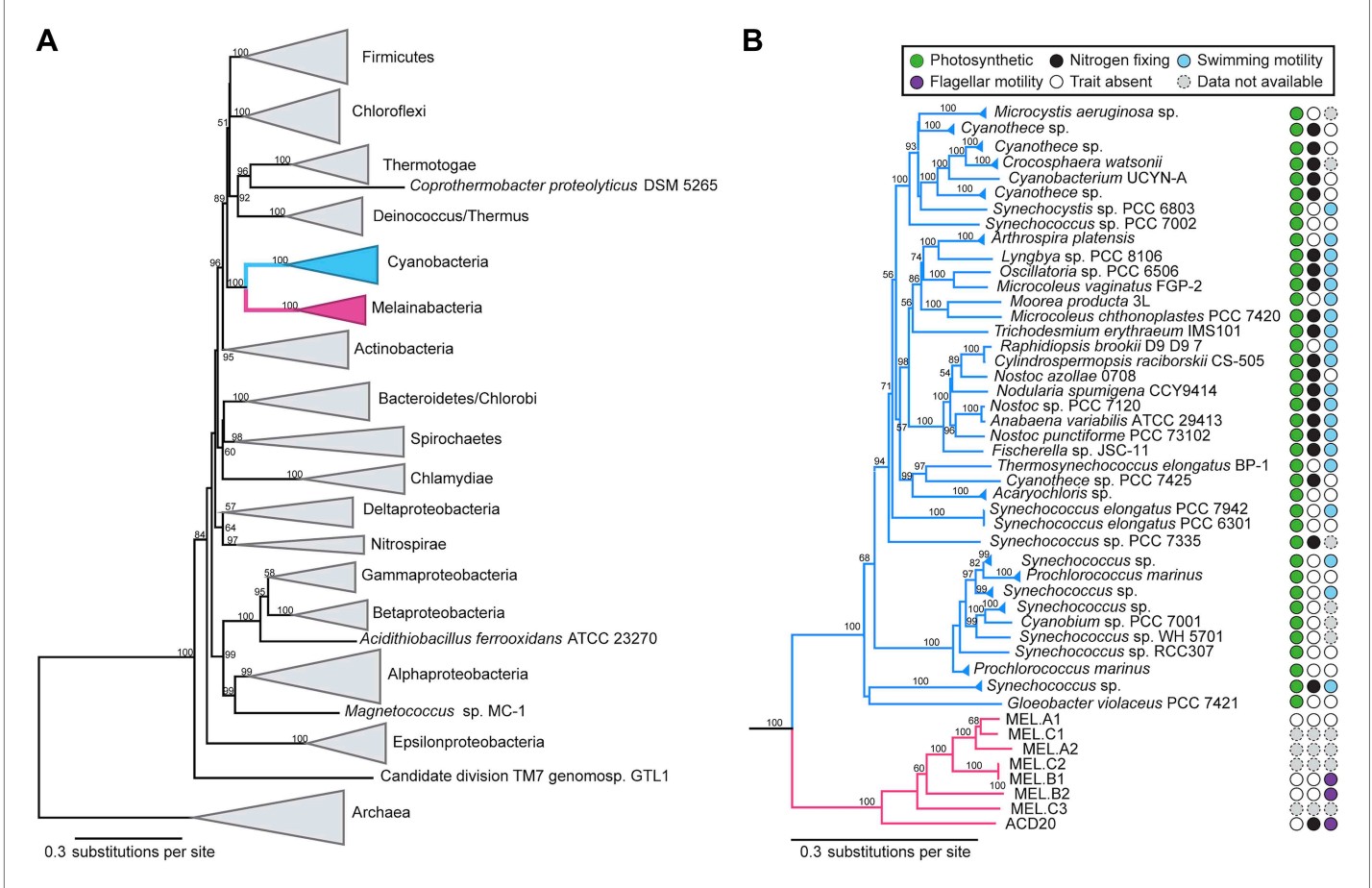

**Figure 3**. Concatenated ribosomal protein phylogeny of the Melainabacteria and Cyanobacteria. Maximum likelihood phylogeny and trait-based comparison of the eight novel organisms and 80 Cyanobacteria based on a concatenated protein alignment of 16 core ribosomal proteins from 733 taxa. In (**A**) the complete tree is shown at the phylum level and in (**B**) only the cyanobacterial-melainabacterial portion of the tree is shown. Bootstrap values >50% are indicated. Cyanobacteria branches are colored blue and Melainabacteria branches, red. The complete tree with all taxa shown is provided in *Figure 3—figure supplement 1*. The protein alignment on which this tree is based is provided in *Figure 3—source data 1*.

The following source data and figure supplements are available for figure 3:

**Source data 1**. Concatenated protein alignment of 16 core ribosomal proteins from 733 taxa and the eight Melainabacteria described here.

**Figure supplement 1**. Complete phylogeny of 733 taxa and the eight Melainabacteria based on a concatenated protein alignment of 16 core ribosomal proteins.

RpaB is likely also exclusive to Cyanobacteria; however, given this gene's similarity to other response regulators it could be considered present in two other bacterial genomes, *Variovorax paradoxus* and *Desulfomicrobium baculatum*. In all cases, the three Melainabacteria genes have highest homology to those present in Cyanobacteria (*Supplementary file 1*).

## Melainabacteria are encapsulated by a Cyanobacteria-like cell envelope

Like Cyanobacteria, Melainabacteria are inferred to have a Gram-negative cell envelope. This conclusion is based on the large number of genes for lipopolysaccharide (LPS), Lipid A biosynthesis, and O-antigen polymerases and transporters found in the curated genomes described here (*Supplementary file 1*). LPS, Lipid A, and O-antigen are components of the Gram-negative outer membrane; in contrast, Gram-positive bacteria lack an outer membrane and therefore lack these structures (*Purves et al., 2003*). This finding is significant, because many genes in the newly reconstructed genomes share closest sequence similarity to Gram-positive members of the Firmicutes (see below). As previously

mentioned, the genomes encode a homolog to an S-layer-like protein, suggesting that the cell envelope has an S-layer. S-layers and O-antigens have both been previously observed on photosynthetic Cyanobacteria (*Hoiczyk and Hansel, 2000*). Taken together, the cell envelope is likely similar to that of Cyanobacteria, consistent with a shared vertical ancestry.

## Melainabacteria are obligate anaerobic fermenters

Based on the lack of a linked electron transport chain, aerobic or anaerobic respiratory complexes (see above), and a complete TCA cycle (see below), and the presence of fermentative and degradative enzymes, we infer that Melainabacteria are obligate anaerobic fermenters. We predict that Melainabacteria can use a wide variety of carbon compounds, including hemicellulosic compounds (only ACD20), polysaccharides, oligosaccharides, and simple sugars, as well as organic acids, amino acids, and fatty acids to yield hydrogen, lactate, acetate (ACD20), formate (Gut), hydrogen, possibly butyrate (ACD20) and ethanol (Gut) (*Figure 4*). Specific sugars predicted to be fermented are glucose, fructose, sorbitol, mannose, trehalose, starch, glycogen, hemicellulose, and amylose, and the relevant enzymes are α-galactosidase, β-galactosidase, α-glucosidase, β-glucosidase, β-glucuronidase, β-fructofranosidase (sucrase), α-mannosidase, pullulanase, α-amylase, and endo-1, 4-beta-xylanase. These enzymes facilitate the utilization of a variety of sugar compounds by degrading the compounds into simpler sugars that can enter the main Embden-Meyerhof-Parnas (EMP) glycolytic pathway. This pathway contains not the classical ATP-dependent enzyme, but a pyrophosphate-dependent phosphofructokinase, a gene found in diverse organisms capable of anaerobic glycolysis (*Mertens, 1991*). This difference in phosphoryl donor specificity may confer an energetic advantage to the cell when glycolysis is the primary source of ATP (*Figure 4*, gene 3). The genomes have the genes necessary for hexose interconversion in the EMP pathway via the pentose phosphate pathway, such that ribose, arabinose, xylulose, and other five-carbon sugar or sugar-alcohols may be utilized. Unlike most Cyanobacteria, which use internal carbon pools for fermentation (*Stal and Moezelaar, 1997*), Melainabacteria likely acquire sugars and sugar-alcohols from the environment via a variety of cytoplasmic membrane permeases (*Figure 4*). Overall, the cells are inferred to have a lifestyle analogous to anaerobic obligate fermentative bacteria known to play an important role in carbon transformation in gut (*Mackie, 2002*) and subsurface systems.

## Metabolic differences between Melainabacteria and Cyanobacteria are represented by genes scattered throughout the Melainabacteria genome

Nearly 30% of the 920 core conserved orthologous genes (COGs) match most closely to genes belonging to members of the phylum Firmicutes that have a fermentative-based metabolism, compared to 15% with closest matches to cyanobacterial genes (*Figure 5A*). These non-cyanobacterial genes are spread throughout the genomes (*Figure 5—figure supplement 1*), arguing against acquisition via a recent lateral transfer event or a chimeric assembly artifact. The COGs whose best match are to COGs of Firmicutes are enriched in functions related to metabolism (*Figure 5B*), including metabolism of carbohydrates, amino sugars, nucleotides, amino acids, and vitamins. It should be noted that while the best match to these genes are within the Firmicutes phylum, the abundance of Firmicutes genomes in databases may have inherently biased this result. These results corroborate extensive divergence in the metabolic lifestyles of Cyanobacteria and Melainabacteria.

## The unusual hydrogenases of the Melainabacteria suggest a syntrophic H$_2$−producing niche

Both gut and subsurface Melainabacteria studied here have trimeric confurcating FeFe hydrogenases with motifs required for catalytic activity (*Figure 4*, *Supplementary file 1*). One of the three confurcating hydrogenases in ACD20, ACD20_18461, has all the necessary residues for functionality (L1, L2, and L3 motifs), while the other two, ACD20_9246_G0007 and ACD20_9246_G0010, contain all three motifs but have a replacement of a serine by cysteine in motif 1 (TSCSPGW rather than TSCCPAW) and have replaced the cysteine in motif 3 with isoleucine. The gut Melainabacteria FeFe hydrogenase has a complete L1 motif but lacks L2 and L3 motifs. These hydrogenases may indicate an ecological role in H$_2$ production in both gut and subsurface systems: syntenic homologs for the three subunits have been identified in genomes of obligate syntrophs and fermenters known to produce high molar ratios of H$_2$ (*Sieber et al., 2012*). The production of H$_2$ typically requires a syntrophic association with

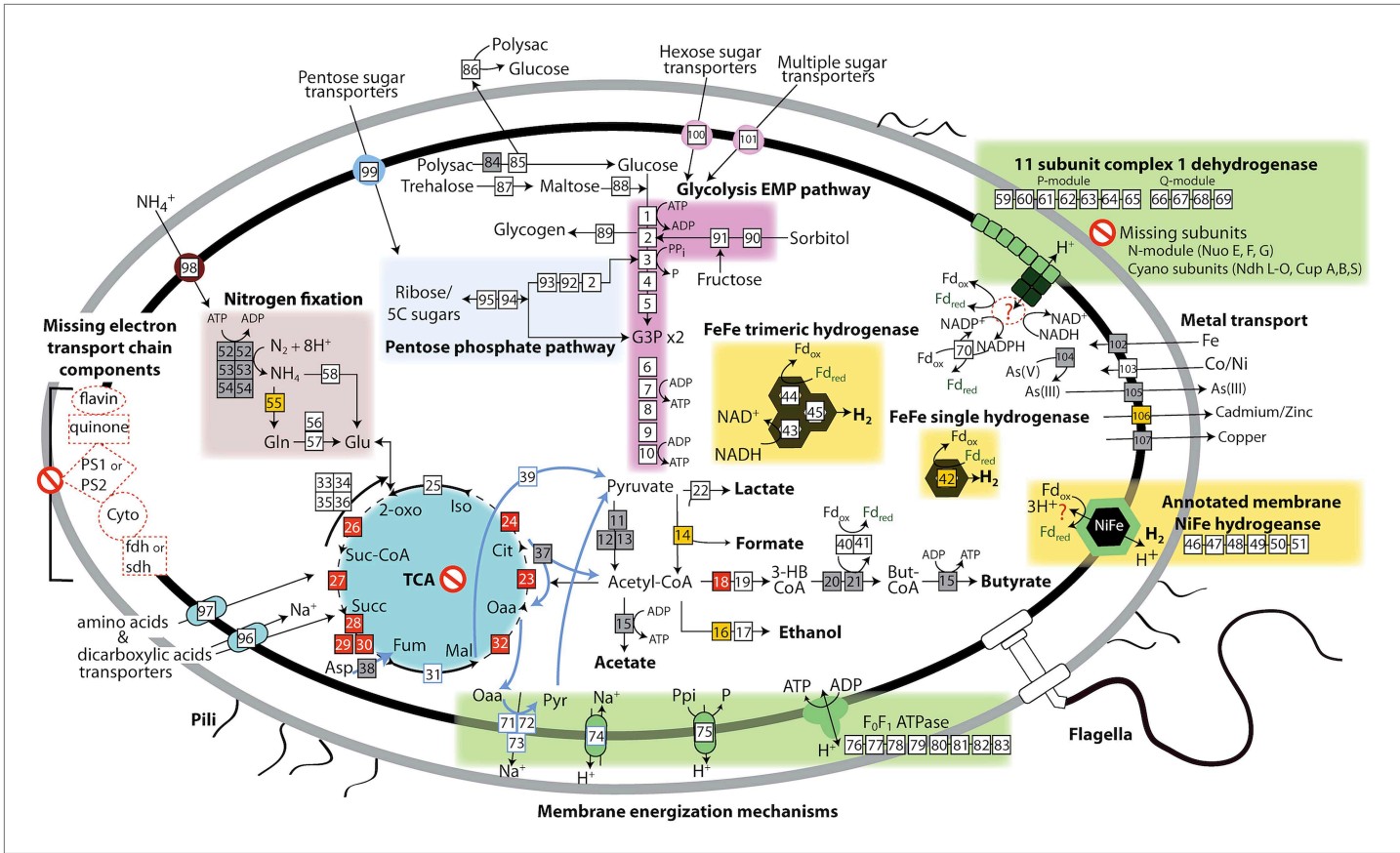

**Figure 4**. The physiological and metabolic landscape of Melainabacteria. Metabolic predictions for Melainabacteria based on genes identified in *Figure 4—source data 1*. Genes in pathways detected in the genomes of the subsurface and at least one gut genome (white box), only in the subsurface genome (grey box), only in at least one gut genome (orange box), and genes missing from pathways in all genomes (red box). Glycolysis proceeds via the canonical Embden-Meyerhof-Parnas (EMP) pathway with the exception of fructose-6-phosphate 1-phosphotransferase (EC:2.7.1.90, gene 3). Names of pathways and fermentation end-products are bolded and ATP generated by substrate-level phosphorylation are noted. All Melainabacteria genomes sampled lack electron transport chain components (including cytochromes (Cyto), succinate dehydrogenase (sdh), flavins, quinones), terminal respiratory oxidases or reductases, and photosystem I or II (PS1, PS2). The genomes also lack a complete TCA cycle (absent enzymes noted by red boxes), with the TCA enzymes instead linked to the fermentation of amino acids and organic acids denoted (pathways, blue arrows). Ferredoxin (Fd, green text) is important for hydrogen ($H_2$) production via hydrogenases (yellow background box). Proton translocation mechanisms (green background box) may be achieved by the activity of trimeric oxaloacetate (OAA) decarboxylase and sodium-hydrogen antiporter, pyrophosphate (PPi) hydrolysis with pyrophosphatases, 11 subunit NADH dehydrogenase, and an annotated NiFe hydrogenase (green enzyme). Annotations for the gene numbers are in *Figure 4—source data 2*. The complete metabolic comparison of the Melainabacteria can be accessed at http://ggkbase.berkeley.edu/genome_summaries/81-MEL-Metabolic-Overview-June2013.

The following source data are available for figure 4:

**Source data 1**. Examination of enzymes (steps) in near-complete KEGG based modules shared among or unique to subsurface ACD20 and gut Melainabacteria genomes MEL.A1, MEL.B1, and MEL.B2.

**Source data 2**. Gene annotations corresponding to the numbers in *Figure 4*.

an $H_2$-consuming partner to maintain low partial hydrogen pressures. Therefore, in addition to being anaerobic fermenters, the Melainabacteria may be $H_2$ producers living in syntrophy with archaeal methanogens or bacterial acetogens in the human gut and with respiring organisms in the subsurface.

## The Melainabacteria genomes encode an incomplete TCA cycle

With *Prochlorococcus*, marine *Synechococcus*, and UCYN-A as exceptions, the vast majority of Cyanobacteria have a complete TCA cycle (*Zhang and Bryant, 2011*). In contrast, the Melainabacteria genomes reported here encode no more than four unlinked genes from the TCA and reverse TCA

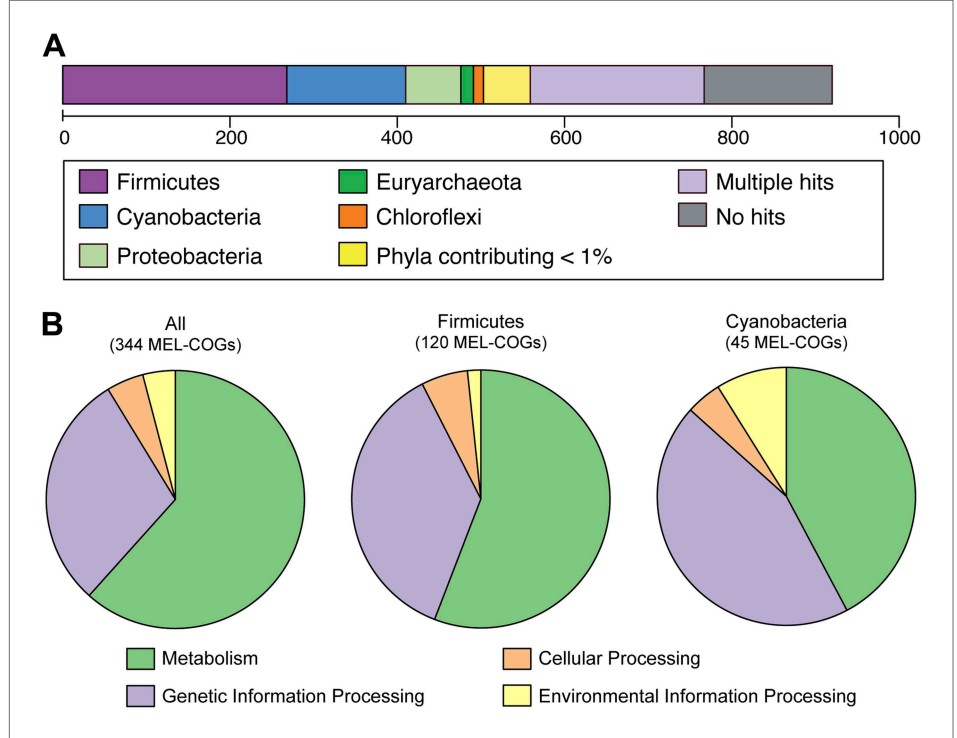

**Figure 5**. MEL-COG phylum and functional assignments. (**A**) Assignment of the 920 MEL-COGs (***Figure 5—source data 1***) to their best matching phyla. (**B**) Functional assignment of COGs by phylum assignment. Only COGs with functional assignments were considered. Number of MEL-COGs with no/multiple functional assignments are 532/42, 136/12, and 87/10 for All, Firmicutes, and Cyanobacteria, respectively. DOI: 10.7554/eLife.01102.013

The following source data and figure supplements are available for figure 5:

**Source data 1**. List of 920 MEL-COGs, including their assigned phylum and KEGG Orthology (KO) identifier. DOI: 10.7554/eLife.01102.014

**Figure supplement 1**. Distribution of MEL-COGs with best hits from different phyla across the MEL-A1 genome. DOI: 10.7554/eLife.01102.015

cycles (***Figure 4***). We also confirmed that the genomes lack alternative enzymes first identified in *Synechococcus* species PCC 7002 that functionally complete the TCA cycle (***Zhang and Bryant, 2011***). Moreover, the genomes have an NAD+ not the NADP+ dependent isocitrate dehydrogenase found in most Cyanobacteria. The absence of a complete TCA cycle necessitates an external requirement for dicarboxylic acids, which could be imported by the two dicarboxylic acid transporters found in each genome. This observation, along with the need for an H$_2$-consuming partner, could guide development of growth media for future isolation efforts.

We infer that the TCA cycle enzymes in the genomes reported here function to link to nitrogen metabolism and energy generating pathways. Both the ACD20 and MEL.B2 genomes encode isocitrate dehydrogenase and fumarase, whose end-products are important intermediates in nitrogen assimilation and amino acid pathways. Unlike the genomes of the gut bacteria, ACD20 can augment glycolytic ATP substrate-level phosphorylation with additional ATP generation using TCA cycle intermediates. For example, unidirectional aspartate ammonia-lyase, fumarase, and malic enzyme can convert amino acids (alanine and aspartate) and organic acids to pyruvate and ultimately acetate with ATP generation. Also, like citrate fermentation in *Klebsiella pneuemoniae* (***Dimroth, 1980***; ***Dimroth and Schink, 1998***), ACD20 may use citrate lyase in conjunction with the combined action of membrane-bound oxaloacetate decarboxylase (EC 4.1.1.3) to pump sodium ions. Combined with a sodium-hydrogen antiporter, a hydrogen gradient can be generated and used to drive cellular processes (***Figure 4***, blue arrows). Hence, the TCA cycle does not appear to function as it does in most Cyanobacteria, where it generates reducing equivalents for oxidative phosphorylation, but rather links to nitrogen metabolism and organic acid fermentation for energy generation.

## Production of proton-motive force and $H_2$

In the absence of a complete electron transport chain (above), the gut and aquifer bacteria studied here appear to augment the ATP produced by substrate-level phosphorylation by membrane energization. This membrane potential can be capitalized on for ATP synthesis by the F-type ATP synthase, used for flagellar motion (see below), or used for solute transport. Given that the human gut associated Melainabacteria only produce ATP by substrate-level phosphorylation in glycolysis, these alternative modes of generating an energized membrane may be important to their overall energy balance.

The Melainabacteria can generate a membrane potential by four mechanisms: (i) a sodium gradient generated by decarboxylation of oxaloacetate (using an oxaloacetate decarboxylase and sodium-hydrogen antiporter, see TCA cycle); (ii) membrane bound $H^+$-translocating pyrophosphatases, which use some of the energy liberated during inorganic pyrophosphate (PPi) hydrolysis to drive proton translocation across the cytoplasmic membrane (*Schocke and Schink, 1998*); (iii) an 11-subunit complex I dehydrogenase (*Battchikova et al., 2011*); and (iv) a putatively annotated six subunit NiFe membrane-bound hydrogenase that lacks required hydrogen binding motifs (*Vignais and Billoud, 2007*; *Marreiros et al., 2013*). Notably, these genomes all lack membrane-associated *Rhodobacter* nitrogen fixation (Rnf) and formate dehydrogenase complexes found in the genomes of obligate fermentative organisms (*McInerney et al., 2007*; *Biegel et al., 2011*). Given the high demand for reduced ferredoxin in the cell, we have considered that both the complex I and the annotated NiFe hydrogenase may use the proton-motive force to produce reduced ferredoxin, which is a required electron donor for the FeFe hydrogenase and nitrogenase systems.

## The gut Melainabacteria may provide their host with B and K vitamins

The Melainabacteria genomes encode complete pathways for biosynthesis of vitamins B2 (riboflavin), B3 (nicotinamide), B7 (biotin), and B9 (dihydrofolate). The gut types additionally make vitamin B5 (pantoate). We are unsure if the subsurface bacterium ACD20 can make vitamin B5 as it appears to lack the final enzyme required in the synthesis of vitamin B5, 2-dehydropantoate 2-reductase. ACD20 and one of the human gut types (MEL.A1) may also be able to synthesize vitamins K1 and K2. Cyanobacteria are capable of synthesizing the B and K vitamins as well (*Kanehisa et al., 2012*). Germ-free animals raised aseptically, and which lack gut microbiota, have an increased nutrient requirement for B and K vitamins, suggesting that under normal conditions the mammalian gut microbiota are a source of these vitamins for the host (*Backhed et al., 2005*). Hence, Melainabacteria may represent one of the microbial sources of the B and K vitamins for their hosts.

## ACD20 is capable of nitrogen fixation by a nitrogenase complex distinct from that in Cyanobacteria

Nitrogen fixation is a capacity common among Cyanobacteria and is accomplished via a nitrogenase complex (*Zehr et al., 2003*). While the human gut-derived melainabacterial genomes lack the genes required for a functional nitrogenase complex (*nifD, nifK, and nifH*), the ACD20 genome encodes these genes and the *nifE, nifV, nifS, nifU, nifB, and nifB/X* genes involved in nitrogen fixation. We confirmed that the ACD20 NifH protein sequence contains the required [4Fe/4S] cluster, all motifs for functionality, and a conserved lysine in position 15 responsible for ATP interaction. Therefore, it seems likely that ACD20 has the capacity to fix nitrogen. This ability has been proposed to account for the increased dominance of *Geobacter* species under ammonium limiting conditions created during acetate stimulated U(VI) bioremediation of the same subsurface aquifer (*Mouser et al., 2009*). The intestinal relatives do not have nitrogenase capabilities. (*Figure 4*, *Supplementary file 1*).

When placed in phylogenetic context with 865 *nifH* gene sequences (*Zehr et al., 2003*), the *nifH* gene from ACD20 (*Figure 6*, red) does not cluster with the primary cyanobacterial *nifH* genes (*Figure 6*, green) in *nifH* group I, but is affiliated with *nifH* group III. Group III is composed of sequences from phylogenetically distant organisms, many of which are obligate anaerobes (e.g., *Clostridium* species, sulfate-reducers, and methanogens) (*Zehr et al., 2003*) (*Figure 6*) as well as *nifH* from some Cyanobacteria (e.g., *Microcoleus chthonoplastes* PCC7420 and *Anabaena variabillis* ATCC 29413, which have secondary copies of *nifH*; *Figure 6*, shown in green in group III). The phylogenetic placement within the *nifH* cluster III was robust to alignment curation method (manual or automatic with GBLOCKS), and the ACD20 *nifH* sequence was never monophyletic with cyanobacterial sequences. The best hit to the ACD20 *nifH* sequence is a group III nitrogenase annotated from *Methanoregula*

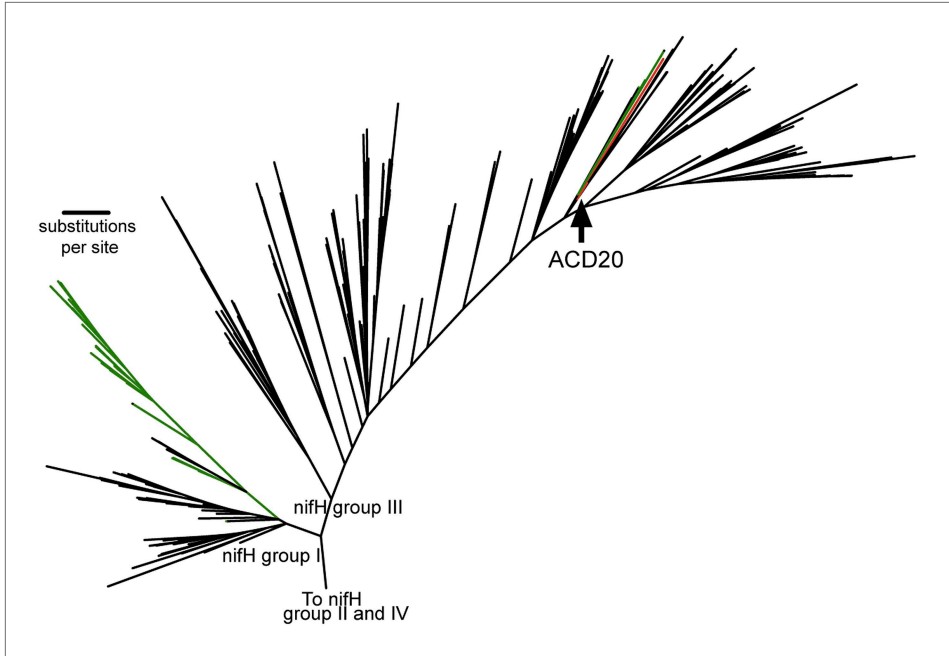

**Figure 6**. The phylogeny of the Melainabacteria nitrogenase. A maximum likelihood phylogenetic tree constructed with 865 nitrogenase *nifH* genes from sequenced genomes (*Zehr et al., 2003*) is shown. *nifH* groups I and III are shown. The ACD20 *nifH* (in group III) is denoted in red, while photosynthetic cyanobacterial *nifH* sequences (in groups I and III) are denoted in green. Relative to group I, group III is characterized by deep bifurcations and long-branch lengths (*Zehr et al., 2003*), which are represented in the constructed tree by low-bootstrap values (<50) for internal branch positions in group III. ACD20 sequences are monophyletic (but with low bootstrap support) with *nifH* sequences from anaerobic *Clostridium* and *Fusobacterium* species.

*boonei* 6A8 (ABS56522). These results indicate that the melainabacterial nitrogenase is not related to the primary nitrogenase in Cyanobacteria.

Further distinctions between the ACD20 and cyanobacterial nitrogen metabolism include how nitrogen is assimilated. The large and small subunit of the glutamine oxoglutarate aminotransferase (GOGAT) system, responsible for nitrogen assimilation in Melainabacteria, are NADP+-based. This finding distinguishes Melainabacteria from known Cyanobacteria, which use either a 3Fe-4S ferredoxin-dependent monomeric enzyme or a two subunit NADH-dependent GOGAT (*Muro-Pastor et al., 2005*). While a common nitrogenase could have existed in the ancestor of Melainabacteria and Cyanobacteria, the extant capacity to fix and assimilate nitrogen appears to have been acquired independently in Cyanobacteria and this sibling lineage.

## Melainabacteria are flagellated

Unlike Cyanobacteria, the organisms studied here are flagellated. All four of the analyzed genomes contain genes for flagella production, and all but one of the analyzed gut genomes (MEL.A1) contains the requisite genes to produce a fully functional flagellum (*Supplementary file 1*). The flagella are composed of the M, S, P, and L rings, as expected given a Gram-negative cell envelope (*DePamphilis and Adler, 1971*). At least one copy of the flagellin protein in each Melainabacteria genome contains the eight amino acid sequence recognized by Toll-like receptor 5 (*Andersen-Nissen et al., 2005*), indicating an ability to interact with the host immune system (*Figure 7*). Trees built using flagellum-related gene sequences show that the genes branch deeply with the Firmicutes and Spirochaetes (*Figure 8*), arguably the two most basal bacterial lineages (*Daubin et al., 2002*; *Ciccarelli et al., 2006*). This result suggests that the common ancestor of Cyanobacteria and Melainabacteria may have been flagellated. Given that no flagellated Cyanobacteria are known and cyanobacterial motility is accomplished by gliding or twitching (*Schaechter, 2010*), flagella may have been non-essential to the cyanobacterial lifestyle and hence lost. However, it is not possible to rule out the alternative of flagella being acquired by Melainabacteria after the divergence from Cyanobacteria.

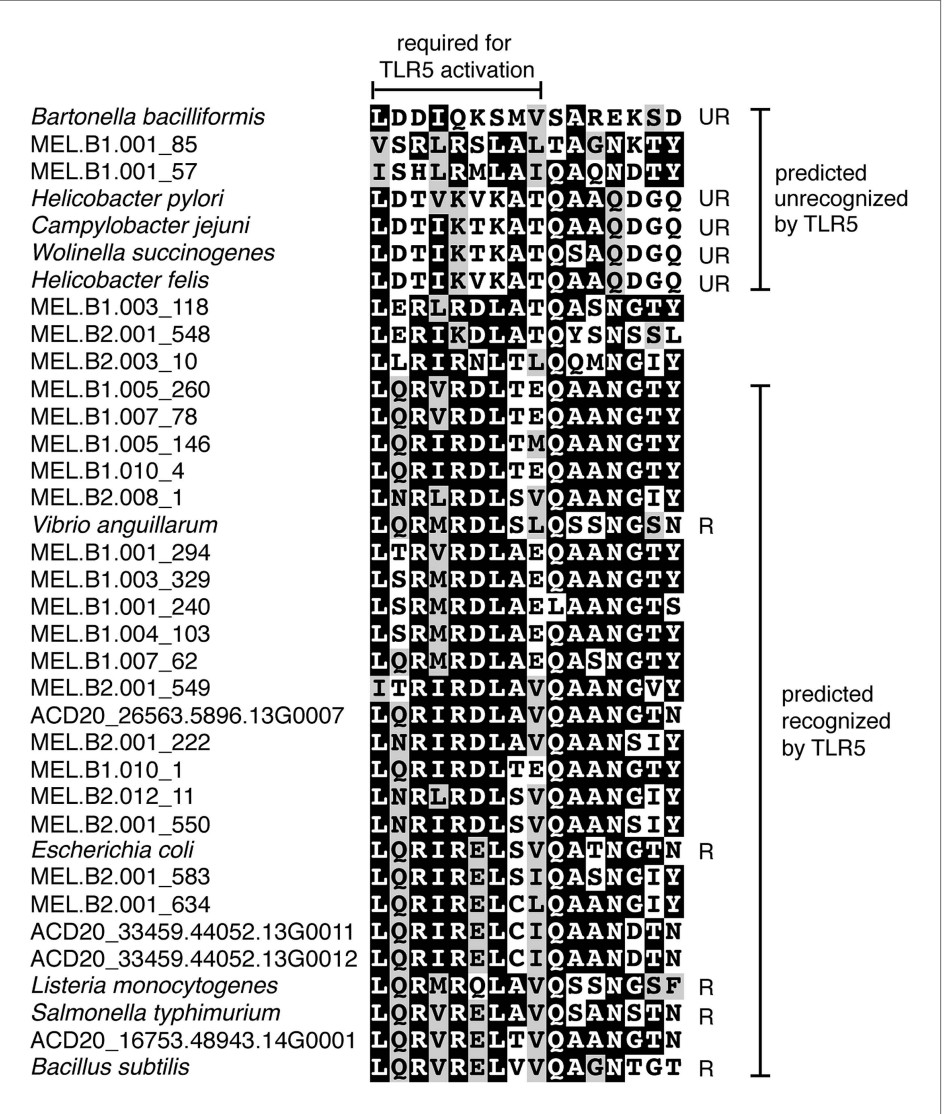

**Figure 7**. Putative TLR5 activation region in Melainabacteria flagellin genes. Protein sequence alignment of residues 88–103 (*Escherichia coli* coordinates) for the flagellin genes. The range of residues required for TLR5 activation (***Andersen-Nissen et al., 2005***) are indicated by the top bracket. Sequences are organized by similarity within these residues. Species whose flagellin are reported (***Andersen-Nissen et al., 2005***) to be recognized (R) or unrecognized (UR) by TLR5 are noted. Based on the visualization of the alignment, flagellin genes predicted to be recognized or unrecognized by TLR5 are indicated; genes of ambiguous TLR5 recognition status are unmarked.

## The gut Melainabacteria have reduced genomes

The gut–associated clade of the Melainabacteria, which diverged from the sediment-associated lineage, appears to have undergone genome reduction relative to ACD20 (***Figure 3***, ***Table 2***). The gut types lack genes for chemotaxis, production of some amino acids (aspartic acid, asparagine, phenylalanine, arginine, histidine, tyrosine, and, in MEL.A1 and MEL.B1, alanine), a type I secretion system, nitrogen fixation, and genes for additional energy generation by substrate-level phosphorylation and for the production of acetate and perhaps butyrate as fermentation end-products. Note that in ACD20, genes for four of the five steps in butyrate synthesis were identified. The fifth step, which could be carried out by butyrate kinase and phosphotransbutyrylase or butyryl-coenzyme A (CoA): acetate CoA-transferase (as occurs in anaerobic bacteria) was not detected. An alternative enzyme, analogous to the process used by some Archaea (e.g., *Pyrococcus* species), acetate CoA ligase (***Mai and Adams, 1996***), found in ACD20, may substitute to convert butyryl CoA to butyrate.

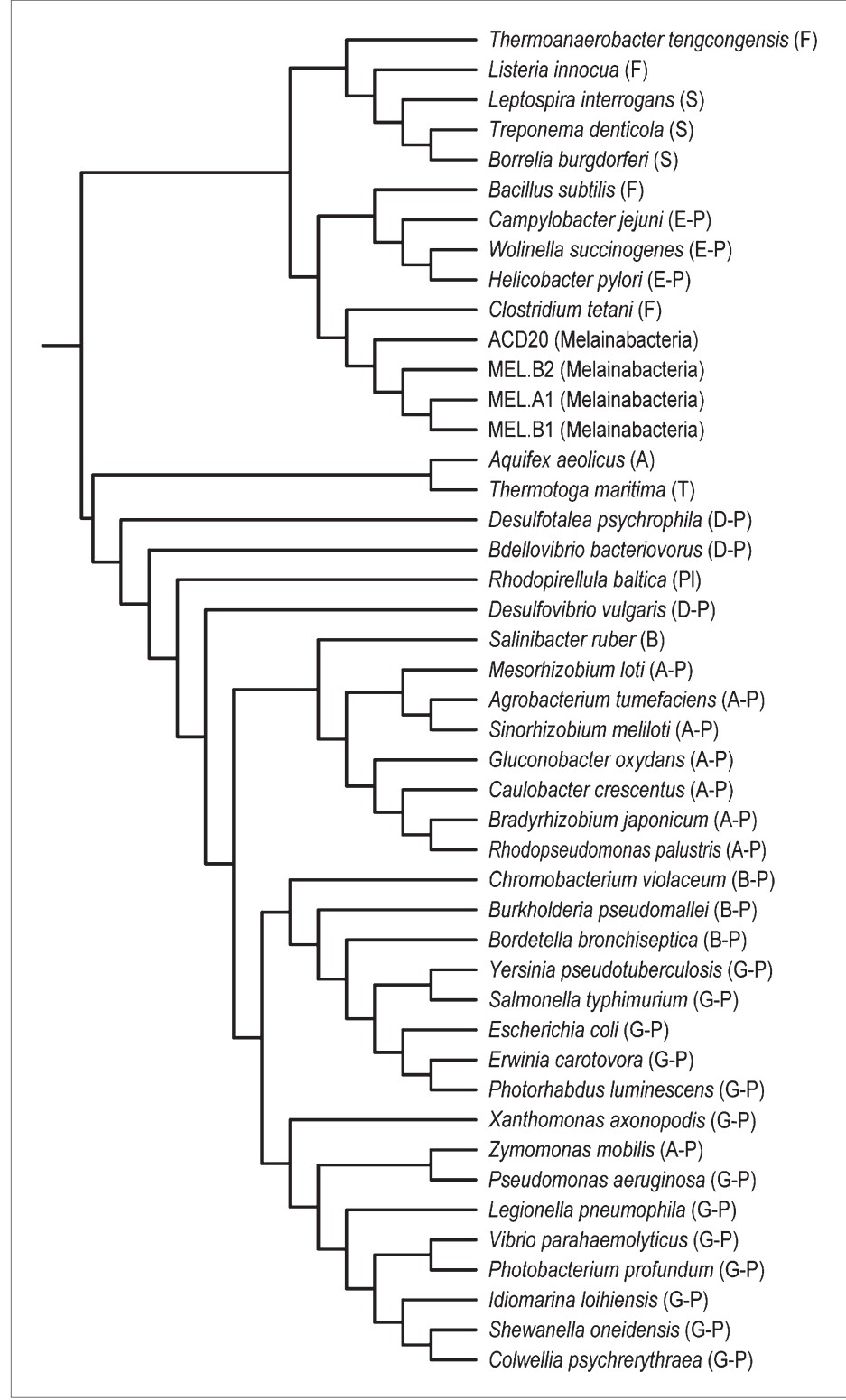

**Figure 8**. Phylogeny of flagella-related genes. Supertree (cladogram) of 13 bootstrap ML trees of the flagellar genes shared among the four analyzed genomes. The phylum (or more specific taxonomic identifier) of each species is listed: (F) Firmicutes, (S) Spirochaetes, (E-P) Epsilonproteobacteria, (MEL) Melainabacteria, (A) Aquificae, (T) Thermotogae, (D-P) Deltaproteobacteria, (Pl) Planctomycetes, (B) Bacteroidetes, (A-P) Alphaproteobacteria, (B-P) Betaproteobacteria, (G-P) Gammaproteobacteria. In all 13 individual trees, Melainabacteria branched with Firmicutes and Spirochaetes.

Genome reduction has been observed in symbiotic bacteria and may have occurred in Melainabacteria during adaptation to life in the animal gut. The lack of the filament, filament cap, hook-filament junction, and L and P rings required for flagellum biosynthesis in the genome assembled into a single scaffold (MEL.A1) indicates that these genes were likely not missed in the genome sequencing and assembly but rather that further genome reduction may be ongoing in the gut-associated clade.

## Distribution of Melainabacteria in soil, water, and animal habitats

To update our view of the ecological niches for Melainabacteria, we searched for 16S rRNA gene sequences in recent datasets (*Figure 9—source data 1*). We detected the water-soil-sediment (non-animal associated) clade in a wide variety of environments (*Figure 9*), with the highest abundances in municipal water. An analysis of the Human Microbiome Project (*Consortium HMP, 2012*) and other datasets (*Figure 9—source data 1*) showed that across the human body, they occur as rare members of skin, airway, and mouth communities, but are most abundant in fecal samples (*Figure 9B*). The gut types were also detected in fecal samples from a range of other mammalian species, with the highest levels in herbivores, consistent with a role in fermentation of dietary substrates (*Figure 9C*). Within the herbivores, Melainabacteria are more abundant in feces obtained from foregut than hindgut fermenters (two tailed *t*-test, p=0.026). Within the Human Microbiome Project dataset (*Consortium HMP, 2012*), approximately 10% of the samples contained melainabacterial 16S rRNA gene sequences, providing a rough estimate as to what fraction of the American population carries Melainabacteria. When comparing three global human populations (*Yatsunenko et al., 2012*), we observed the highest abundances in fecal samples obtained from predominantly vegetarian Malawian and Venezuelan individuals (*Figure 9C*). These observations suggest that Melainabacteria also play a role in fermentation of dietary plant polysaccharides in humans.

## Discussion

High throughput sequencing and bioinformatic methods enabled recovery of four complete and four partial genomes for relatively rare members of complex microbial communities from the adult human gut and groundwater. Our approach used standard metagenomic sequencing and assembly approaches, augmented by manual curation. Recovery of complete genomes was facilitated by the uniqueness of the Melainabacteria genomes, which were distinct not only from the genomes of all other organisms in the community but also were differentiable among coexisting Melainabacteria. This approach has the potential to replace culture-based, single cell, and flow cytometry-based genomics methods, which are costlier and more laborious due to the requirement for cell manipulation. Moreover, the amplification step in single cell genomics often results in partial genomes (*Marcy et al., 2007*; *Podar et al., 2007*; *Youssef et al., 2011*) and chimeras (*Zhang et al., 2006*).

Traditionally, novel bacterial 16S rRNA sequences have been assigned to new candidate phyla based on low levels of sequence identity to existing phyla, and use of this criterion supports the assignment of these bacteria to a new phylum distinct from Cyanobacteria (*Hugenholtz et al., 1998*); here we suggest reclassification of sequences previously referred to as deep branching Cyanobacteria, YS2, SM1D11, or mle1-12 to the candidate phylum Melainabacteria. Similarly, ACD20 should be assigned to the Melainabacteria candidate phylum. Both 16S rRNA and concatenated ribosomal protein phylogenetic trees support common ancestry for the Melainabacteria and the Cyanobacteria lineages. Melainabacteria are, therefore, not only important members of the human gut but their unique evolutionary placement also provides novel insight into the evolution of Cyanobacteria.

Our metabolic reconstruction indicates that the Melainabacteria are anaerobic, obligate fermenters capable of utilizing diverse carbon sources, which are imported as well as derived from internal carbon sources. Melainabacteria lack a linked electron transport chain but have multiple methods for generating a membrane potential, which can then produce ATP via an ATP synthase. Distinct from Cyanobacteria (which have NiFe hydrogenases [*Tamagnini et al., 2007*]), the Melainabacteria cells use FeFe hydrogenases for $H_2$ production and may require an $H_2$-sink (e.g., a coexisting acetogen, methanogen, or other respiring partner) to maintain low partial $H_2$ pressures.

The metabolic profile and nutritional requirements of Melainabacteria uncovered here provide guidance into how these bacteria may be cultured. Efforts to culture Melainabacteria should focus on providing a diverse array of sugars and carbohydrates, dicarboxylic acids, and a complete set of amino acids in an anaerobic, dark environment. Moreover, given their production of $H_2$, these strategies should include removal of hydrogen through use of an $H_2$-consuming syntrophic partner.

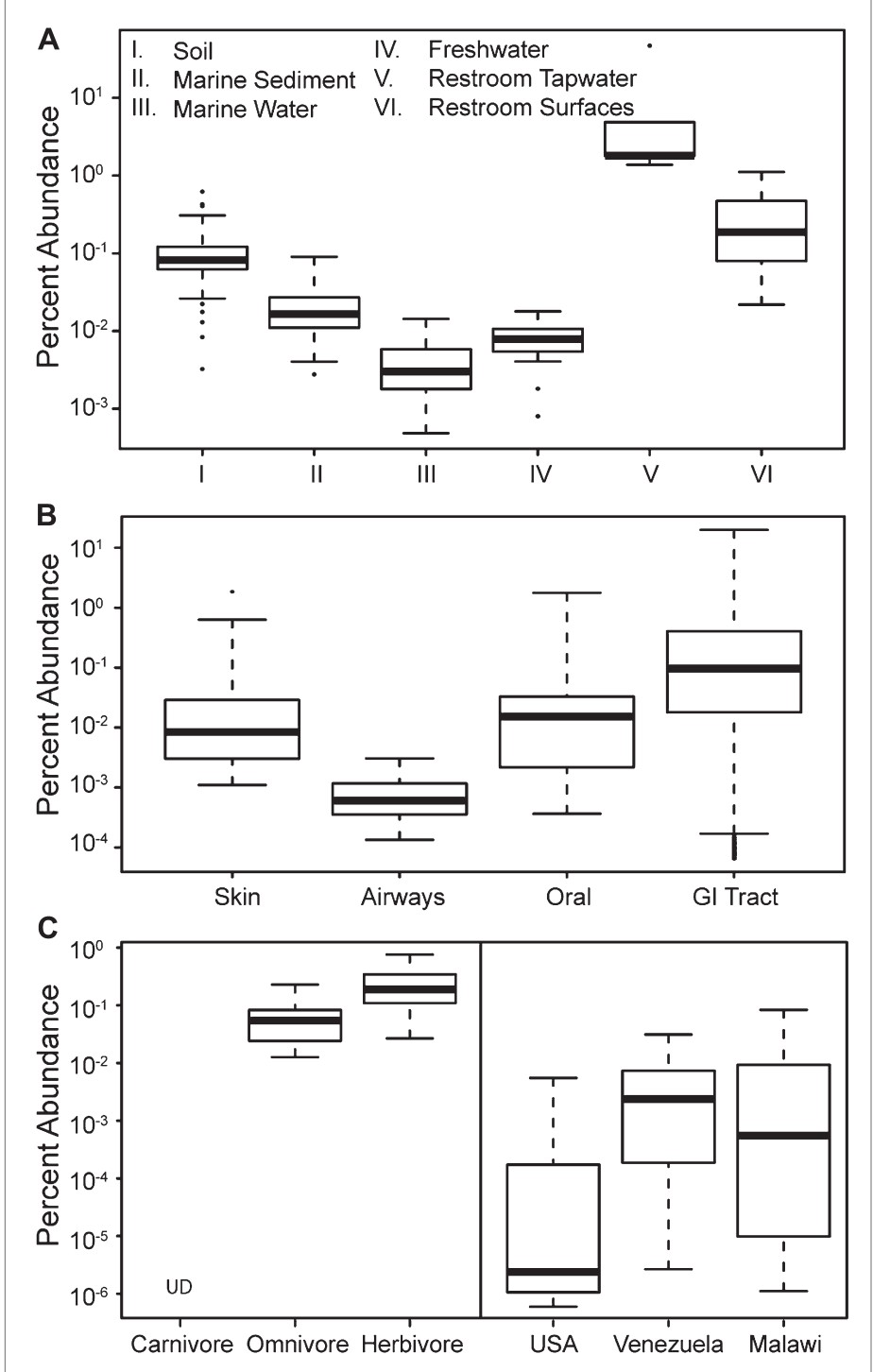

**Figure 9**. The prevalence of members of Melainabacteria in different environments, including distinct human body habitats. In all three panels, the relative abundances of Melainabacteria in different samples types are plotted as box plots (log$_{10}$ transformed; i.e., $10^{-1}$ = 0.1%). Data were obtained from the QIIME database, derive from a variety of studies, and are publically available (*Figure 9—source data 1*): (**A**) soil, sediment, and water sites, (**B**) different human body sites (GI = gastrointestinal), (**C**, left) mammal stool classified by host diet, (**C**, right) country of origin for human stool. UD = undetermined.

The following source data are available for figure 9:

**Source data 1**. 16S rRNA gene sequence datasets used to analyze the sources of Melainabacteria.

We infer that the protein sequences of the flagellin genes of Melainabacteria are recognized by Toll-like receptor 5 (as are flagellins of some commensal Firmicutes in the mammalian gut) that mediates interaction with the host. The human gut Melainabacteria may provide their host with B and K vitamins and fill a niche based on carbon fermentation in the large intestine, analogous to that of the Firmicutes. Based on their distribution in animals and humans living on different continents and consuming different diets, we hypothesize that the gut Melainabacteria flourish in the presence of plant polysaccharides and may aid in the digestion of plant fibers.

Oxygenic photosynthesis, a unique form of photosynthesis that originated in the cyanobacterial lineage, was one of the most profound innovations of life on Earth as it induced planetary-scale geochemical changes and facilitated the evolution of plants via endosymbiosis of cyanobacterial cells (*Sagan, 1967*; *Bekker et al., 2004*; *Mulkidjanian et al., 2006*). We infer that this capacity did not arise prior to the divergence of Melainabacteria from Cyanobacteria, based on the absence of remnants of photosynthetic machinery in the Melainabacteria genomes. This observation is in contrast to the symbiotic photoheterotroph UCYN-A, which is the sole representative of Cyanobacteria unable to perform oxygenic photosynthesis due to loss of photosystem II (*Tripp et al., 2010*). A similar loss of photosynthesis appears to have occurred in the nitrogen-fixing endosymbiont of the algae *Rhopalodia gibba* (*Kneip et al., 2008*). The presence of a light stress response regulator (NblS) and circadian rhythm regulator (RpaA and RpaB) homologs in this sibling phylum suggests that the cyanobacterial ability to sense and use light was preceded and perhaps facilitated by response regulator systems present in the proto-Cyanobacteria.

The timing of the evolution of nitrogen fixation in Cyanobacteria has been debated (*Sanchez-Baracaldo et al., 2005*; *Mulkidjanian et al., 2006*; *Shi and Falkowski, 2008*; *Larsson et al., 2011*; *Latysheva et al., 2012*). Like Cyanobacteria, the subsurface Melainabacteria (ACD20) is capable of nitrogen fixation but its nitrogenase genes appear to be unrelated. Though we cannot rule out the existence of an ancient nitrogenase common in both lineages, this finding suggests that acquisition of nitrogen fixation occurred after Cyanobacteria and Melainabacteria diverged. This sequence of events is in agreement with other theories, which place the development of photosynthesis before nitrogen fixation (*Sanchez-Baracaldo et al., 2005*; *Shi and Falkowski, 2008*). Thus, we speculate that Cyanobacteria gained the capacities for oxygenic photosynthesis and nitrogen fixation and lost or never gained flagella after diverging from Melainabacteria. Melainabacteria on the other hand, gained or maintained flagella, a fermentation system, and an array of carbon transporters. One environmental (non-gut) lineage of Melainabacteria acquired nitrogen fixation, whereas the gut lineage capitalized on its ability to ferment diverse carbon sources, including those recalcitrant to host digestion.

We conclude that the common ancestors of Cyanobacteria and Melainabacteria may have been Gram-negative flagellated bacteria active in the anaerobic carbon cycle, producing $H_2$, but not fixing nitrogen. Although this characterization does not clarify the origins of photosynthesis, it is consistent with the idea that the ancestors of Cyanobacteria were anoxygenic photosynthetic organisms with a photosystem I-like reaction center (*Mulkidjanian et al., 2006*). The inferred lifestyle is consistent with the widely accepted hypothesis that the common ancestor of extant bacteria relied on anaerobic decomposition of organic material by substrate-level phosphorylation (*Egami, 1977*) and was likely metabolically similar to the Firmicutes (*Daubin et al., 2002*; *Ciccarelli et al., 2006*).

## Prospectus

The reconstruction of genomes from uncultivated bacteria from human gut and subsurface environmental metagenomes has enabled us to describe Melainabacteria, a novel candidate phylum sibling to Cyanobacteria, and to further elucidate the evolutionary history of one of the Earth's most important bacterial phyla. Our results suggest that while photosynthesis probably developed in Cyanobacteria after the separation from Melainabacteria, the evolution of light related capabilities may have been enabled by regulators present in their common ancestor. The ability to fix nitrogen appears to have developed separately in Cyanobacteria and the water-soil-sediment clade of Melainabacteria. The role of the Melainabacteria in the human gut is one of an obligate fermenter, and its enrichment in human subjects and animals with a plant-rich diet likely relates to a prominent role of the Melainabacteria in the processing of plant fibers. As plant fibers have been minimized in Western diets, the Melainabacteria may be regarded as part of the microbiota that is disappearing from modernized populations (*Blaser and Falkow, 2009*). The metabolic reconstructions made from these genomes should guide efforts into obtaining Melainabacteria in culture, which would allow a better understanding of this important human symbiont.

## Materials and methods

### Human gut sample collection

The three fecal samples from healthy adults (A, B, and C) were collected under Cornell University IRB (Protocol ID 1108002388) from the United Kingdom Adult Twin Registry (TwinsUK). Samples were collected in 15 ml conical tubes, refrigerated for 1–2 days, and then stored at −80°C at King's College London until being shipped on dry ice to Cornell University, where they were subsequently stored at −80°C. Approximately 100 mg of sample was processed with PowerSoil DNA isolation kit (MoBio Laboratories Ltd, Carlsbad, CA) to isolate genomic DNA.

### Fecal sample DNA sequencing

Construction of three shotgun genomic libraries and sequencing were carried out at the WM Keck Center for Comparative and Functional Genomics, Roy J Carver Biotechnology Center, University of Illinois at Urbana-Champaign. The barcoded DNAseq libraries were prepared with Illumina's 'TruSeq DNAseq Sample Prep kit' (Illumina, San Diego, CA). The final libraries were quantitated with Qubit (Life Technologies, Grand Island, NY), and the average size was determined on an Agilent bioanalyzer High-Sensitivity DNA chip (Agilent Technologies, Wilmington, DE), diluted to 10 nM and pooled. The 10 nM dilution was further quantitated by qPCR on an ABI 7900 (Life Technologies).

The pooled libraries were sequenced on one lane of a flowcell for 101 cycles from each end of the fragments on a HiSeq2000 using TruSeq SBS sequencing kit version 3, and the fastq files were generated with Casava1.8.2. Overall, sequencing yielded a total of 346 million reads (34.6 Gbp, almost 12 Gbp per sample on average).

### Aquifer sample collection and sequencing

Details relating to the collection, sequencing, and data analysis for the ACD20 genome are provided in (*Wrighton et al., 2012*). This publication focused on members of the OP11, OD1, PER, ACD80, and BD1-5 candidate bacterial phyla and included no discussion of the ACD20 genome.

### Genome assembly

We reconstructed four complete (MEL.A1, MEL.B1, MEL.B2, and MEL.C1), one near-complete (MEL. C2), and two partial (MEL.A2 and MEL.C3) genomes from the three human fecal samples. An eighth near-complete genome (ACD20) was recovered from a different dataset (see *Wrighton et al. (2012)* for genome assembly details). Potential metagenomic sequences belonging to Melainabacteria were determined by similarity to Cyanobacteria or by being completely novel. These scaffold fragments were linked to other fragments by coverage, %GC, and paired-end read information using the assembly curation steps and scripts previously described (*Sharon et al., 2013*). Genome MEL.A1 was assembled from human fecal sample A using Velvet (*Zerbino and Birney, 2008*) with parameters optimized based on the expected genome coverage. Genome MEL.A2 was recovered from an IDBA-UD (*Peng et al., 2012*) assembly for human fecal sample A based on a phylogenetic profile of hits for the scaffolds and genes. Genomes MEL.B1 and MEL.B2 were reconstructed from human fecal sample B and genomes MEL.C1, MEL.C2, and MEL.C3 from human fecal sample C. These genomes were assembled using the IDBA-UD assembler. Identification of scaffolds belonging to the target Melainabacteria genome was aided by utilizing similarity to genes in MEL.A1. Scaffolds not belonging to MEL.C1 or MEL.C2 were identified as belonging to genome MEL.C3 when more than 50% of the best hits for its genes were to genes in the other Melainabacteria genomes and not from other published genomes.

Genome completeness was assessed as follows: Complete genomes have a complete set of single copy genes (*Raes et al., 2007*) (*Figure 10*), and the linkage between all scaffolds is established; Near complete genomes have a complete set of single copy genes (*Figure 10*), and the linkage between almost all scaffolds is established; Partial genomes lack a complete set of single copy genes, and/or scaffold linkage is lacking. The newly sequenced gut genomes are available at http://ggkbase.berkeley.edu/mel/organisms.

### Concatenated ribosomal protein phylogeny

A core group of 16 syntenic ribosomal proteins was selected based on published metrics of lateral gene transfer frequencies (*rpL2, 3, 4, 5, 6, 14, 15, 16, 18, 22, 24,* and *rpS3, 8, 10, 17, 19*) (*Sorek et al., 2007*; *Wu and Eisen, 2008*). Reference datasets were obtained from the PhyloSift database

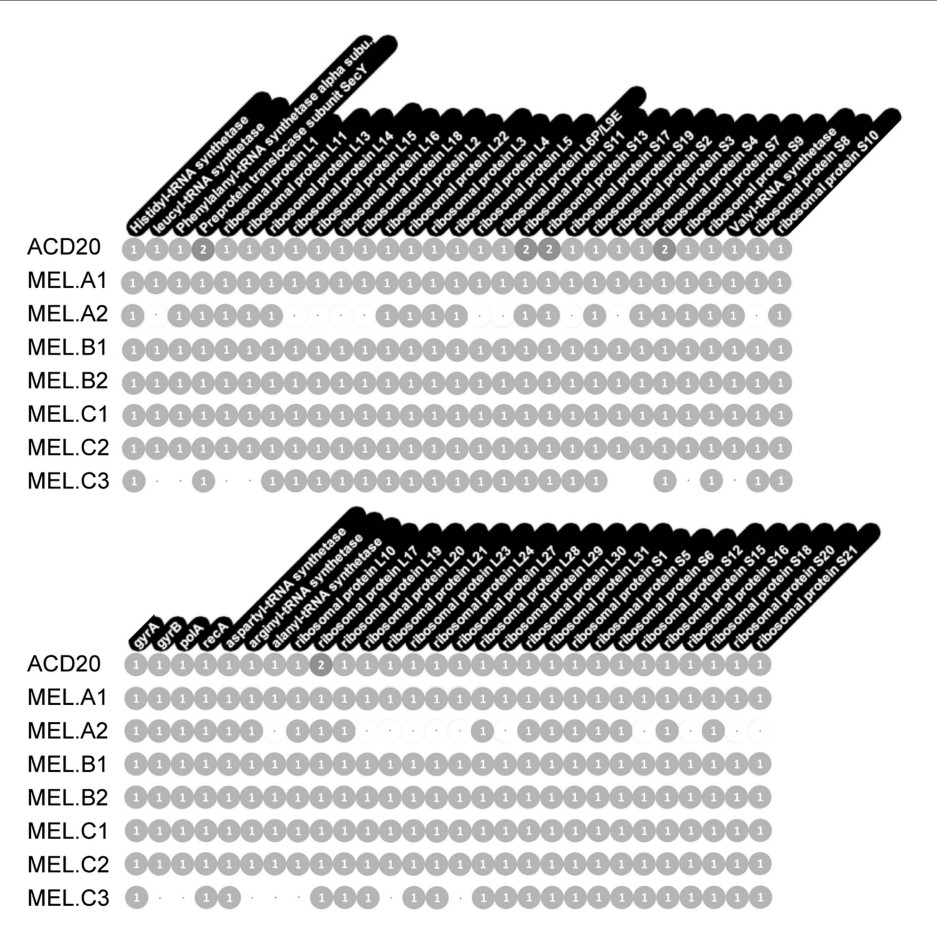

**Figure 10**. Single copy gene inventory from reconstructed genomes. Data are based on single copy genes (numbers in circles indicate the number of copies found).

(*Darling et al., 2012*). The NCBI and JGI IMG databases were mined for the 16 ribosomal proteins from recently sequenced genomes from the Cyanobacteria, Chloroflexi, Nitrospira, and TM7 phyla. The 16 syntenic ribosomal protein genes were identified in the eight new genomes, translated, and included for phylogenetic placement. The complete dataset contained 733 taxa. Each individual protein dataset was aligned using Muscle version 3.8.31 (*Edgar, 2004*) and then manually curated to remove end gaps and single-taxon insertions. Model selection for evolutionary analysis was determined using ProtTest3 (*Darriba et al., 2011*) for each single protein alignment. The curated alignments were concatenated to form a 16-protein, 733 taxa alignment with 3,082 unambiguously aligned positions. A maximum likelihood phylogeny for the concatenated alignment was conducted using Phyml (*Guindon and Gascuel, 2003*) under the LG+αG model of evolution and with 100 bootstrap replicates.

## 16S rRNA gene phylogeny

16S rRNA gene sequences from MEL.A1, MEL.B1, and MEL.B2 were aligned using NAST (*McDonald et al., 2012*). An aligned full-length sequence set was created for five representative sequences from each phylum in the Greengenes (May 2011) database (*DeSantis et al., 2006*) plus the three Melainabacteria sequences. For the Cyanobacteria-Melainabacteria specific tree, the aligned sequence set was composed of the three Melainabacteria sequences and one representative sequence from each order within Cyanobacteria except for orders YS2, SM1D11, and mle1-12, from which we included all sequences as these are most closely related to the Melainabacteria sequences. Phylogenetic trees were constructed using maximum likelihood implemented in RAxML (*Stamatakis, 2006*) and using the

GTR+γ+I model of evolution and 100 bootstrap resamplings. The trees were rooted using five archaeal sequences. Trees were visualized using the Interactive Tree of Life (iTOL) (*Letunic and Bork, 2007*, *2011*).

## RnpB structure analysis

RnpB RNAs were identified by searching for matches to 5′ GAGGAAAGUCC 3′, which is highly conserved in RNase P RNAs (*Haas et al., 1991*), as well as the surrounding intergenic region. Matches were analyzed in Bcheck (*Yusuf et al., 2010*) to determine the structure and type of the RnpB RNAs.

## Choice of genomes used for metabolic analyses

The following four genomes were used for the metabolic analyses: ACD20, MEL.A1, MEL.B1, and MEL.B2. Two partial genomes (MEL.A2 and MEL.C3) were excluded from this analysis because of reliance on clusters of orthologs from all participating genomes. MEL.C1 and MEL.C2 were not included in these analyses because of their similarity to MEL.A1 and MEL.B1, respectively.

## Construction of Melainabacteria clusters of orthologous genes (MEL-COGs)

Clusters of orthologs (920 MEL-COGs in total, *Figure 5—source data 1*) for four of the Melainabacteria genomes (ACD20, MEL.A1, MEL.B1, and MEL.B2) were constructed by (1) BLASTing each protein from each genome against the proteome of the other genomes individually, (2) connecting proteins between pairs of genomes by reciprocal best BLAST hits, and (3) forming clusters of these reciprocal best hits when groups of four proteins (one from each genome) had at least two proteins connected to all three other members of the group, and the rest were connected to at least two other members of the group. The COGs were annotated with (I) phylogenetic origin according to best matching phylum in a BLAST search of NCBI's nr database and (II) KEGG category according to the BLAST match of the ACD20 homolog in the KEGG database.

## Phylogenetic assignments of MEL-COGs

Phylogenetic assignments were decided based on majority voting for all four genes of the MEL-COG (one gene from each of the four Melainabacteria genomes analyzed [ACD20, MEL.A1, MEL.B1, and MEL.B2]). First, a phylum was assigned for each single gene. Then, the phylum assigned to majority of the four MEL-COG genes was assigned to the MEL-COG. If none of the MEL-COG genes received a significant hit to an existing gene in another phylum, the MEL-COG was assigned 'No hit'. If more than one phylum shared the highest number of assignments, or several MEL-COG genes had a 'Multiple' assignment from #3 below, the MEL-COG was assigned 'Multiple'.

Phylum assignment to each individual gene in a MEL-COG was accomplished using homology to genes belonging to other phyla using the following workflow: (1) MEL-COG genes were BLASTed against NCBI's nr database; (2) The top two hits with e-value ≤ 1e-5 were collected; and (3) The alignments of these genes with the MEL-COG gene were compared, considering both alignment-length and % identity: each alignment was represented by the total number of identical positions, calculated as (alignment-length) × (% identity). The top hit was considered to be a significantly better match if its number of identical positions was at least 5% more than the number of identical positions for the second best matching gene. In that case, the phylum of the best hit was assigned to the MEL-COG gene; otherwise the MEL-COG gene received the 'Multiple' assignment.

## KEGG category assignment of MEL-COGs

The ACD20 representative in each MEL-COG was BLASTed against the Kyoto Encyclopedia of Genes and Genomes (KEGG) database (5). The KEGG term for the best hit was assigned to the MEL-COG. If no hits were found, the MEL-COG assignment was 'unknown', else all pathways to which the KEGG term belongs were identified, as well as their KEGG category. Assignment was: (a) 'Undecided'—if the KEGG term did not belong to any pathway; (b) 'Multiple'—if the pathways to which the KEGG term belongs were from multiple categories; or, (c) the category of the term's pathway, if all pathways belonged to the same category.

## Metabolic pathways

KEGG annotations for the ACD20, MEL.A1, MEL.B1, and MEL.B2 genomes were uploaded to the KEGG database. Pathways were visually inspected for completeness, reactants, and products. All annotations were confirmed by manual inspection, including confirmation of active residues and phylogenetic tree analyses. Phylogenetic trees were constructed using protein alignments as described in detail (*Wrighton et al., 2012*). Confirmation and cross-genome comparisons were constructed by using the LIST and GENOME

SUMMARY features in the ggKbase website (http://ggkbase.berkeley.edu/genome_summaries/81-MEL-Metabolic-Overview-June2013). Genomic information from the Melainabacteria dataset is stored in the publically accessible ggKbase database (http://ggkbase.berkeley.edu/mel/organisms).

## Nitrogenase complex phylogenetic analysis

Phylogenetic analyses of the *nifH* gene constructed with a database of 865 *nifH* sequences from genome-sequenced Bacteria and Archaea (*Zehr et al., 2003*). Two separate phylogenetic trees were constructed with and without the use of GBLOCKS in the pipeline reported previously (*Wrighton et al., 2012*). *Figure 6* is based upon the GBLOCKS alignment. Maximum likelihood phylogenetic trees were produced using RAxML version 7.3.0 using the rapid bootstrap analysis and the general time reversible model of nucleotide substitution with optimization and categorization of per-site substitution rates on 500 distinct trees (raxmlHPC -f a -m GTRCAT -x 1234 -N 500) (*Stamatakis, 2006*) and visualized in iTOL (*Letunic and Bork, 2007*, *2011*).

## Flagellum-related gene set phylogenetic analysis

Phylogenetic trees were built from the 13 flagellum-related genes in the MEL-COG list (the ACD20 homolog is listed): ACD20_20398.28785.13G0015 (*flgG*), ACD20_20398.28785.13G0016 (*flgG*), ACD20_20398.28785.13G0018 (*flgC*), ACD20_20398.28785.13G0021 (*fliF*), ACD20_20398.28785.13G0023 (*fliH*), ACD20_20398.28785.13G0024 (*flhA*), ACD20_20398.28785.13G0025 (*fliI*), ACD20_20398.28785.13G0028 (*flgE*), ACD20_26563.5896.13G0002 (*flhA*), ACD20_26723.8006.14G0006 (*fliP*), ACD20_26723.8006.14G0007 (*fliQ*), ACD20_26723.8006.14G0009 (*fliR*), ACD20_29089.28969.14G0027 (*flhB*). For each gene, the ACD20 homolog was used in a pBLAST search (e-value <10$^{-5}$) against the 41 flagellated bacterial species used in (*Liu and Ochman, 2007*). Matching protein sequences across the queried species were aligned using MUSCLE version 3.6 (*Edgar, 2004*). Protein alignments were converted to DNA alignments. Phylogenetic trees were produced by maximum likelihood using RAxML version 7.3.0, executing the rapid bootstrap analysis and the general time reversible model of nucleotide substitution with optimization and categorization of per-site substitution rates (50 rate categories) on 1000 distinct trees (raxmlHPC -e 0.1 -f a -c 50 -m GTRCAT -x 92957 -N 1000 -p 3212) (*Stamatakis, 2006*). A supertree was built using heuristic searches in Clann (*Creevey and McInerney, 2005*).

## Flagellin and Toll-like receptor 5

Flagellin genes were recovered from the ACD20, MEL.A1, MEL.B1, and MEL.B2 genomes. The flagellin gene MEL.B1.001_31 was excluded as this sequence is incomplete and does not align well to other flagellin sequences. Using MUSCLE (3.8) multiple sequence alignment (*Edgar, 2004*), these sequences were aligned with representative flagellin genes from species whose flagellins are known to be either recognized or unrecognized by the mammalian Toll-like receptor 5 (*Andersen-Nissen et al., 2005*) (gene names and Uniprot IDs are given): *E. coli fliC* (Q0GJI9), *Bacillus subtilis hag* (P02968), *Salmonella* Typhimurium *fliC* (P06179), *Vibrio anguillarum flaC* (Q56574), *Listeria monocytogenes flaA* (Q02551), *Bartonella bacilliformis fla1* (P35633), *Campylobacter jejuni flaA* (Q46113), *Helicobacter pylori flaA* (P0A0S1), *Helicobacter felis flaA* (Q9XB38), *Wolinella succinogenes flaG* (Q79HP6). For visualization purposes, the sequences were ordered by similarity within the TLR5 activation domain (*Andersen-Nissen et al., 2005*). BOXSHADE (http://www.ch.embnet.org/software/BOX_form.html) was used to display the protein alignment.

## Meta-analysis of publically available 16S rRNA data

We built a reference dataset of environmental (non-gut) and gut associated Melainabacteria 16S rRNA gene sequences (*Figure 2*) obtained from the Greengenes database (*DeSantis et al., 2006*) and queried the dataset at 97% ID against publicly available 16S rRNA gene datasets obtained from http://www.microbio.me/qiime (*Figure 9—source data 1*). Similar samples types were combined (e.g., forest soil, grassland soil, shrubland combined as 'soil'), and Melainabacteria sequence reads were tallied. Samples with zero Melainabacteria sequence reads were removed. Samples types (i.e., air) with fewer than five reads were not plotted.

## Acknowledgements

We would like to thank Norman Pace and Kirk Harris for providing guidance on analyses of the 30S ribosomal protein S1 and of RnpB, Jonathan Eisen for making the PhyloSift database sequences available, Jonathan Zehr and Kendra Turk for providing the *nifH* reference sequences, and Roberto Kolter, Jonathan Zehr, and the anonymous reviewer for their comments on the manuscript.

## Additional information

### Funding

| Funder | Grant reference number | Author |
|---|---|---|
| National Institutes of Health | R01 DK093595 | Ruth E Ley |
| David and Lucile Packard Foundation | 2010-35960 | Ruth E Ley |
| The Hartwell Foundation | | Ruth E Ley |
| Arnold and Mabel Beckman Foundation | | Ruth E Ley |
| DOE IFRC, Subsurface Biogeochemical Research Program, Office of Science, Biological and Environmental Research | DE-AC02-05CH11231 | Jillian F Banfield |
| EMBO | | Itai Sharon |
| Wellcome Trust | | Jordana T Bell, Timothy D Spector |
| National Institute for Health Research | | Jordana T Bell, Timothy D Spector |
| DOE Knowledgebase Program | DE-SC0004918 | Jillian F Banfield |

The funders had no role in study design, data collection and interpretation, or the decision to submit the work for publication.

### Author contributions

SCD, IS, KCW, OK, LAH, JKG, Conception and design, Acquisition of data, Analysis and interpretation of data, Drafting or revising the article; BCT, JTB, TDS, Contributed resources and tools; JFB, REL, Conception and design, Analysis and interpretation of data, Drafting or revising the article

### Ethics

Human subjects: Informed consent and consent to publish was obtained for the human subjects in the TwinsUK project. Ethical approval was obtained and guidelines were followed in accordance with Cornell University IRB (Protocol ID 1108002388).

## Additional files

### Supplementary files

• Supplementary file 1. Genes belonging to pathways or assemblages referenced in the paper.

### Major dataset

The following dataset was generated:

| Author(s) | Year | Dataset title | Dataset ID and/or URL | Database, license, and accessibility information |
|---|---|---|---|---|
| Di Rienzi SC, Sharon I, Wrighton KC, Koren O, Hug LA, Thomas BC, Goodrich JK, Bell JT, Spector TD, Banfield JF, Ley RE | 2013 | The Melainabacteria Project | http://ggkbase.berkeley.edu/mel | Publicly available at ggKBase (http://ggkbase.berkeley.edu/). |

The following previously published dataset was used:

| Author(s) | Year | Dataset title | Dataset ID and/or URL | Database, license, and accessibility information |
|---|---|---|---|---|
| Wrighton KC, Thomas BC, Sharon I, Miller CS, Castelle CJ, Verberkmoes NC, Wilkins MJ, Hettich RL, Lipton MS, Williams KH, Long PE, Banfield JF | 2012 | The ACD Rifle project | http://ggkbase.berkeley.edu/Rifle_ACD | Publicly available at ggKBase (http://ggkbase.berkeley.edu/). |

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
