## [Decision Letter]

[Editors’ note: a previous version of this study was rejected after peer review, but the authors submitted for reconsideration. The two decision letters after peer review are shown below.]

Many thanks for submitting your work on the genome reconstruction and subsequent analyses of the several members of the previously only poor characterized ‘deep branching Cyanobacteria’ (The human gut and subsurface sediment harbor non-photosynthetic Cyanobacteria). In addition to myself (Roberto Kolter), two other individuals read and offered comments on the manuscript. Overall we found the analyses quite intriguing and have discussed our opinions extensively since providing our initial reviews. My sense is that we have come to a good consensus about the manuscript. We all see some important work here but have some serious concerns that can be summarized as follows:

1) There is a major concern that there needs to be better support for the claim that these genomes, as constructed from metagenomic data, do indeed occur in nature (i.e., that they are ‘real’). We see two possibilities here. It may be that you can indeed develop an argument that from the bioinformatic analyses you have absolute confidence that the assembled sequences are true representations of extant genomes in nature. Alternatively, there are experimental ways to determine this. In the spirit of *eLife*, we do not wish to recommend “make work” types of experiments. The former solution should not necessitate bench work; the latter does and could take a lot of time. We recognize that. But you might already have some of those data.

2) A second major concern has to do with the designation of these organisms as Cyanobacteria to begin with. In fact, if anything, we would argue that your results make a very compelling argument that these bacteria need to be considered in a class (phylum?) all by themselves. In this regard, the paper would have to be revised greatly because it currently rests on the assertion that these are “truly non-photosynthetic Cyanobacteria”. This concern is a bit larger than it seems. After all, the main interest in this manuscript was the characterization of non-photosynthetic cyanos. But after reading the results, it is clear that the initial designation of these as Cyanobacteria was probably premature and off the mark. Nonetheless, we feel that this correction should be communicated.

Regarding the overall writing of manuscript (particularly as it pertains to the usage of inappropriate usage of the term ‘Cyanobacteria’) we had split reactions. One reviewer found the paper well written, I must confess that I did not. I marked the PDF extensively with suggested edits.

Considering these major concerns, and the possibility that additional experiments might be needed, I am recommending that the submission be rejected. That will free you and your colleagues to submit the work (perhaps editing it as a results of our comments) elsewhere. However, I want you to consider this a “soft reject” that leaves open the opportunity for a resubmission. If you can make a compelling case that this new phylum is intrinsically important and interesting and can provide a stronger case that the genomes are real, we would be glad to reconsider it. I, for one, would love to see how knowledge of the genomes might guide you and your colleagues to cultivate members of this group. Not that I would hold you to it, but some discussion of why they have not been cultivated based on genome knowledge would be welcome.

[Editors’ note: what now follows is the decision letter after the authors submitted for further consideration.]

Thank you for sending your resubmission to *eLife*, which is now entitled “The human gut and groundwater harbor non-photosynthetic bacteria belonging to a new phylum sibling to Cyanobacteria”. This new version of your article has been favorably evaluated by a Senior editor, myself as a member of the Board of Reviewing Editors, and two expert reviewers (one of them, Jon Zehr agreed to reveal his identity).

After our initial individual evaluation of the manuscript we discussed our comments and reached a consensus that by and large you and your co-authors have addressed our prior main concerns which were: (a) that this division was called “deep branching Cyanobacteria” when all evidence pointed to the bacteria not being cyanos and (b) that there was not enough description of how the genomes had been assembled. In addition, we feel that the revised manuscript is much improved in terms of its written style. Nonetheless, all of us felt that the manuscript could still use some revising before it can be accepted for publication. In short, the key changes still needed include:

1) Given the lack of cultured representatives, this should be still referred to as “candidate phylum”.

2) The discussion of the light sensing genes should be modified and streamlined because the direct evidence that the homologs are indeed involved in light sensing is somewhat weak.

3) The discussion of the evolution of nitrogenase may need to be re-pitched as there is no well-accepted view on the original evolution of nitrogenase. We feel some of your arguments could be used in favor of its being present in the last common ancestor of Cyanobacteria.

---

## [Author Response]

[Editors’ note: the author responses to the first round of peer review follow.]

*1) There is a major concern that there needs to be better support for the claim that these genomes, as constructed from metagenomic data, do indeed occur in nature (i.e., that they are ‘real’). We see two possibilities here. It may be that you can indeed develop an argument that from the bioinformatic analyses you have absolute confidence that the assembled sequences are true representations of extant genomes in nature. Alternatively, there are experimental ways to determine this. In the spirit of eLife, we do not wish to recommend “make work” types of experiments. The former solution should not necessitate bench work; the latter does and could take a lot of time. We recognize that. But you might already have some of those data*.

We have provided a very detailed description of the genome reconstruction process for one of the genomes reported in the study and this process is summarized in the main text of the paper. As you will see, the process is very well controlled and relies on information from multiple independent sources that were used for verifying the assembly. Our criteria for defining complete genomes are very stringent and include far more than the presence of all 57 single copy genes. Specifically, we resolved all connections between assembled scaffolds assigned to the genome. This last criterion makes it very unlikely that parts of the genome will be left out or that foreign fragments will be added to the genome. In fact, given the absence of genome amplification, it is likely that the genomes recovered using these methods are more complete and at least as accurate as genomes recovered by single cell genomics, the widely accepted method for studying uncultivated microbes. Further supporting the veracity of the assemblies, the identification of scaffolds that belong to these genomes was very simple. Beyond this step, the process is very similar to culture-based genomics. Please note that reconstructing genomes from metagenomic data is not a new approach and has a long history in both the Banfield Lab (e.g., Tyson et al., Nature, 2004; Lo et al. Nature, 2007; Aliaga Goltsman et al. Applied and Environmental Microbiology, 2009; Baker et al. PNAS, 2010; Wrighton et al., Science, 2012; Sharon et al., Genome Research, 2013) and other labs (e.g., Iverson et al., Science, 2012; Albertsen et al., Nature Biotechnology, 2013).

Testing the credibility of our methods using, for example, single cell genomics or cultivation is not only difficult but also probably impossible since there is no guarantee that the strain (or species) captured by one of the other methods will be the same as the one assembled from the metagenomic data. However we recently had the opportunity to validate the assembly of a genome recovered by us from subsurface sediment (Castelle et al., Nature Communications, in press) that was assembled using very similar methods to those used here. The validation was done using long reads (∼8 kbp) that were sequenced from the same sample using the Moleculo sequencing technology. 322 of the 340 reads (95%) belonging to the genome aligned perfectly to the assembled genome. Disagreements between the remaining 18 reads and the assembled genome were checked and found to be the result of 11 local mis-assemblies of a few hundred base pairs each (compared to a genome size of 2.2 Mbp). These localized mis-assemblies would also have arisen in an isolate genome assembly. Hence, this experiment provides independent verification of the reliability of our metagenomic genome recovery method.

*2) A second major concern has to do with the designation of these organisms as Cyanobacteria to begin with. In fact, if anything, we would argue that your results make a very compelling argument that these bacteria need to be considered in a class (phylum?) all by themselves. In this regard, the paper would have to be revised greatly because it currently rests on the assertion that these are “truly non-photosythetic Cyanobacteria”. This concern is a bit larger than it seems. After all, the main interest in this manuscript was the characterization of non-photosynthetic cyanos. But after reading the results, it is clear that the initial designation of these as Cyanobacteria was probably premature and off the mark. Nonetheless, we feel that this correction should be communicated*.

We have now revised the manuscript extensively and defined the reported genomes as representing a new phylum. Following a suggestion from one of the reviewers we also propose the name “Melainabacteria” for the new phylum. This decision is supported by commonly accepted criteria for defining new phyla (16S rRNA similarity to known phyla – see text) and we thank the reviewers for raising this important issue. We do insist however that the new genomes are the closest relatives to Cyanobacteria that have been sequenced to date. This relies on both 16S rRNA phylogeny, which is the accepted gold standard for inferring phylogenetic relations, and also phylogeny that is based on concatenated ribosomal proteins, which have proved to be very reliable.

We agree that the message of the paper should not have been restricted to the discovery of “truly non-photosynthetic Cyanobacteria”. We revised the paper to highlight implications for the evolutionary history of Cyanobacteria.

The current study provides a first glimpse into a new group of organisms inhabiting the human gut that is prevalent in the West, appears to be more prevalent in non-Western countries, and may have had an important role in maintaining human health in the past. As an aside, we mention that the group of organisms reported in this study was included in the 200 “most wanted genomes” of the NIH Human Microbiome Project, which underscores the interest of the human microbiome community in these organisms. Taken together, we strongly believe that the results reported here are of interest for a wide range of researchers as expected from a paper submitted to a wide audience journal such as *eLife*.

*[Editors’ note: the author responses to the re-review follow.*]

With regards to the key changes requested:

1) We now refer to the Melainabacteria phylum as a candidate phylum throughout the paper.

2) We have carefully rewritten the light sensing discussion noting that these genes are not capable of sensing light but rather are regulator genes that may have been co-opted for regulating light responsive genes and thus may have aided the acquisition of light systems.

3) We have noted in our discussion of nitrogenase evolution that ambiguity still exists as we cannot exclude the possibility of an ancient shared nitrogenase existing prior to the present nitrogenase complexes in Melainabacteria and Cyanobacteria.

Furthermore, we have also clarified the flagellin TLR5 activation sequence figure (Figure 7) by rearranging the sequences according to similarity in the TLR5 activation region and by indicating which sequences we predict by sequence similarity to be recognized by TLR5. We have also removed the sentence from the Discussion that mentioned our attempts to culture this group. This is because one of us (Ruth) felt that our attempts had been somewhat haphazard, and since culturing anaerobes is not one of our core specialties, this information is not likely to be informative (i.e., another more specialized group may succeed using these methods).